# A chemical probe of CARM1 alters epigenetic plasticity against breast cancer cell invasion

Xiao-Chuan Cai[1], Tuo Zhang[2], Eui-jun Kim[3], Ming Jiang[1,4], Ke Wang[1], Junyi Wang[1], Shi Chen[1,5], Nawei Zhang[1,6], Hong Wu[7], Fengling Li[7], Carlo C dela Seña[7], Hong Zeng[7], Victor Vivcharuk[8], Xiang Niu[9,10], Weihong Zheng[1], Jonghan P Lee[1,5], Yuling Chen[11], Dalia Barsyte[7], Magda Szewczyk[7], Taraneh Hajian[7], Glorymar Ibáñez[1], Aiping Dong[7], Ludmila Dombrovski[7], Zhenyu Zhang[6†], Haiteng Deng[7,11], Jinrong Min[7,12], Cheryl H Arrowsmith[7,13], Linas Mazutis[9], Lei Shi[8], Masoud Vedadi[7,14], Peter J Brown[7], Jenny Xiang[2], Li-Xuan Qin[15], Wei Xu[3], Minkui Luo[1,4]*

[1]Chemical Biology Program, Memorial Sloan Kettering Cancer Center, New York, United States; [2]Genomics Resources Core Facility, Weill Cornell Medical College, Cornell University, New York, United States; [3]McArdle Laboratory for Cancer Research, University of Wisconsin-Madison, Madison, United States; [4]Program of Pharmacology, Weill Cornell Medical College of Cornell University, New York, United States; [5]Tri-Institutional PhD Program in Chemical Biology, Memorial Sloan Kettering Cancer Center, New York, United States; [6]Department of Obstetrics and Gynecology, Chaoyang Hospital, Affiliation Hospital of Capital Medical University, Beijing, China; [7]Structural Genomics Consortium, University of Toronto, Toronto, Canada; [8]Department of Physiology and Biophysics, Weill Cornell Medical College of Cornell University, New York, United States; [9]Computational and Systems Biology Program, Memorial Sloan Kettering Cancer Center, New York, United States; [10]Tri-Institutional PhD Program in Computational Biology and Medicine, Memorial Sloan Kettering Cancer Center, New York, United States; [11]Center for Synthetic and Systematic Biology, School of Life Sciences, Tsinghua University, Beijing, China; [12]Department of Physiology, University of Toronto, Toronto, Canada; [13]Princess Margaret Cancer Centre, Department of Medical Biophysics, University of Toronto, Toronto, Canada; [14]Department of Pharmacology and Toxicology, University of Toronto, Toronto, Canada; [15]Department of Epidemiology and Biostatistics, Memorial Sloan Kettering Cancer Center, New York, United States

*For correspondence:
luom@mskcc.org

†Deceased

Competing interests: The authors declare that no competing interests exist.

**Abstract** CARM1 is a cancer-relevant protein arginine methyltransferase that regulates many aspects of transcription. Its pharmacological inhibition is a promising anti-cancer strategy. Here **SKI-73** (**6a** in this work) is presented as a CARM1 chemical probe with pro-drug properties. **SKI-73** (**6a**) can rapidly penetrate cell membranes and then be processed into active inhibitors, which are retained intracellularly with 10-fold enrichment for several days. These compounds were characterized for their potency, selectivity, modes of **action**, and on-target engagement. **SKI-73** (**6a**) recapitulates the effect of CARM1 knockout against breast cancer cell invasion. Single-cell RNA-seq analysis revealed that the **SKI-73**(**6a**)-associated reduction of invasiveness acts by altering epigenetic plasticity and suppressing the invasion-prone subpopulation. Interestingly, **SKI-73** (**6a**) and CARM1 knockout alter the epigenetic plasticity with remarkable difference, suggesting distinct

modes of action for small-molecule and genetic perturbations. We therefore discovered a CARM1-addiction mechanism of cancer metastasis and developed a chemical probe to target this process.

## Introduction

Numerous biological events are orchestrated epigenetically upon defining cellular fates (*Atlasi and Stunnenberg, 2017*; *Berdasco and Esteller, 2019*). Among the key epigenetic regulators are protein methyltransferases (PMTs), which can render downstream signals by modifying specific Arg or Lys residues of their substrates with *S*-adenosyl-L-methionine (SAM) as a methyl donor cofactor (*Luo, 2018*). Significant efforts have been made to identify the PMT-dependent epigenetic cues that are dysregulated or addicted under specific disease settings such as cancer (*Berdasco and Esteller, 2019*). Many PMTs are implicated as vulnerable targets against cancer malignancy (*Kaniskan et al., 2018*; *Luo, 2018*). The pro-cancerous mechanism of these PMTs can be attributed to their methyltransferase activities, which act individually or in combination to upregulate oncogenes, downregulate tumor suppressors, and maintain cancer-cell-addicted homeostasis (*Berdasco and Esteller, 2019*; *Blanc and Richard, 2017*). Pharmacological inhibition of these epigenetic events thus presents promising anti-cancer strategies (*Berdasco and Esteller, 2019*), as exemplified by the development of the clinical inhibitors of DOT1L (*Bernt et al., 2011*; *Daigle et al., 2011*), EZH2 (*Kim et al., 2013*; *Konze et al., 2013*; *McCabe et al., 2012*; *Qi et al., 2012*; *Qi et al., 2017*), and PRMT5 (*Bonday et al., 2018*; *Chan-Penebre et al., 2015*).

Protein arginine methyltransferases (PRMTs) act on their substrates to yield three different forms of methylated arginine: asymmetric dimethylarginine (ADMA), symmetric dimethylarginine (SDMA), and monomethylarginine (MMA), which are the terminal products of Type I, II and III PRMTs, respectively (*Blanc and Richard, 2017*; *Yang and Bedford, 2013*). Among the important Type I PRMTs is CARM1 (PRMT4), which regulates multiple aspects of transcription by methylating diverse targets including RNAPII, SRC3, C/EBPβ, PAX3/7, SOX2/9, RUNX1, Notch1, p300, CBP, p/CIP, Med12, and BAF155 (*Blanc and Richard, 2017*; *Hein et al., 2015*; *Vu et al., 2013*; *Wang et al., 2015*; *Wang et al., 2014a*; *Yang and Bedford, 2013*). The physiological function of CARM1 has been linked to the differentiation and maturation of embryonic stem cells to form immune cells, adipocytes, chondrocytes, myocytes, and lung tissues (*Blanc and Richard, 2017*; *Yang and Bedford, 2013*). The requirement of CARM1 is implicated in multiple cancers, with its methyltransferase activity particularly addicted by hematopoietic malignancies and metastatic breast cancer (*Drew et al., 2017*; *Greenblatt et al., 2018*; *Nakayama et al., 2018*; *Wang et al., 2014a*). Our prior efforts using in vivo mouse and in vitro cell models uncovered the role of CARM1 in promoting breast cancer metastasis (*Wang et al., 2014a*). Mechanistically, CARM1 methylates Arg1064 of BAF155 and thus facilitates the recruitment of the BAF155-containing SWI/SNF complex to a specific subset of gene loci that are essential for breast cancer metastasis. CARM1 thus emerges as a novel anti-cancer target (*Wang et al., 2014a*).

Although this cancer relevance inspired the development of CARM1 inhibitors (*Kaniskan et al., 2018*; *Scheer et al., 2019*), many small-molecule CARM1 inhibitors lack target selectivity or cellular activity (*Kaniskan et al., 2018*), two essential criteria of chemical probes (*Frye, 2010*). To the best of our knowledge, EZM2302 (*Drew et al., 2017*; *Greenblatt et al., 2018*), TP-064 (*Nakayama et al., 2018*) and **SKI-73** (**6a** in this work, www.thesgc.org/chemical-probes/SKI-73), which were developed by Epizyme, Takeda/SGC(Structural Genomic Consortium), and our team, respectively, are the only selective and cell-active CARM1 chemical probes. EZM2302 and TP-064 were developed from conventional small-molecule scaffolds occupying the substrate-binding pocket of CARM1 (*Drew et al., 2017*; *Greenblatt et al., 2018*; *Nakayama et al., 2018*). The potential utility of EZM2302 and TP-064 is implicated by their selective anti-proliferative effects on hematopoietic cancer cells, in particular multiple myeloma cells (*Drew et al., 2017*; *Greenblatt et al., 2018*; *Nakayama et al., 2018*). However, definitive molecular mechanisms of the CARM1 addiction in these contexts remain elusive (*Greenblatt et al., 2018*).

Here, we report the characterization and novel utility of **SKI-73**, a chemical probe of CARM1 with pro-drug properties. **SKI-73** (**6a** in this work) can readily penetrate cell membranes and then be processed into two active CARM1 inhibitors that contain 6′−homosinefungin (**HSF**) as their core

**eLife digest** Drugs that are small molecules have the potential to block the individual proteins that drive the spread of cancer, but their design is a challenge. This is because they need to get inside the cell and find their target without binding to other proteins on the way. However, small molecule drugs often have an electric charge, which makes it hard for them to cross the cell membrane. Additionally, most proteins are not completely unique, making it harder for the drugs to find the correct target.

CARM1 is a protein that plays a role in the spread of breast cancer cells, and scientists are currently looking for a small molecule that will inhibit its action. The group of enzymes that CARM1 belongs to act by taking a small chemical group, called a methyl group, from a molecule called SAM, and transferring it to proteins that switch genes on and off. In the case of CARM1, this changes cell behavior by turning on genes involved in cell movement. Genetically modifying cells so they will not produce any CARM1 stops the spread of breast cancer cells, but developing a drug with the same effects has proved difficult. Existing drugs that can inhibit CARM1 in a test tube struggle to get inside cells and to distinguish between CARM1 and its related enzymes.

Now, Cai et al. have modified and tested a CARM1 inhibitor to address these problems, and find out how these small molecules work. At its core, the inhibitor has a structure very similar to a SAM molecule, so it can fit into the SAM binding pocket of CARM1 and its related enzymes. To stop the inhibitor from binding to other proteins, Cai et al. made small changes to its structure until it only interacted with CARM1.Then, to get the inhibitor inside breast cancer cells, Cai et al. cloaked its charged area with a chemical shield, allowing it to cross the cell membrane. Inside the cell, the chemical shield broke away, allowing the inhibitor to attach to CARM1. Analysis of cells showed that this inhibition only affected the cancer cells most likely to spread. Blocking CARM1 switched off genes involved in cell movement and stopped cancer cells from travelling through 3D gels.

This work is a step towards making a drug that can block CARM1 in cancer cells, but there is still further work to be done. The next stages will be to test whether the new inhibitor works in other types of cancer cells, in living animals, and in human patient samples.

scaffold (*Scheer et al., 2019*; *Wu et al., 2016*). Notably, the two inhibitors can accumulate inside cells to remarkably high concentrations and for a prolonged period. The potency, selectivity, modes of action, on-target engagement, and off-target effects of these compounds were characterized with multiple orthogonal assays in vitro and under cellular settings. The pharmacological inhibition of CARM1 by **SKI-73** (**6a**) recapitulates the anti-invasion effect of the genetic perturbation of CARM1. In the context of cellular heterogeneity, we developed a cell-cycle-aware algorithm for single-cell RNA-seq (scRNA-seq) analysis and dissected the invasion-prone subset of breast cancer cells that is sensitive to **SKI-73** (**6a**) treatment. Our scRNA-seq analysis provides the unprecedented insight that pharmacological inhibition of CARM1 alters epigenetic plasticity and suppresses invasion by suppressing the most invasive subpopulation of breast cancer cells.

## Results

### Development of 6′−homosinefungin derivatives as potent and selective CARM1 inhibitors

Upon developing cofactor-competitive PMT inhibitors (*Wu et al., 2016*; *Zheng et al., 2012*), we tailored the SAM analog sinefungin (*Figure 1a*) around its 6′-amino moiety to potentially engage CARM1′s substrate-binding pocket. 6′-homosinefungin (**HSF**, i.e., **1**), a sinefungin analog with the insertion of 6′−methylene moiety, was discovered for its general high affinity to Type I PRMTs (*Figure 1a,b*, *Figure 1—figure supplement 1*, *Supplementary file 1*-Table A). As a SAM mimic, **1** binds to the Type I PRMTs (namely PRMT1, CARM1, PRMT6 and PRMT8) with $IC_{50}$ of 13–300 nM (*Figure 1a,c*, *Supplementary file 1*-Table A). Its relative affinity to Type I PRMTs aligns with that of the SAM mimics **SAH** and **SNF** (around 20-fold lower $IC_{50}$ of **1** versus **SAH** and **SNF**, *Figure 1a,c*, *Supplementary file 1*-Table A). This observation argues that **1** retains the structural features of **SAH**

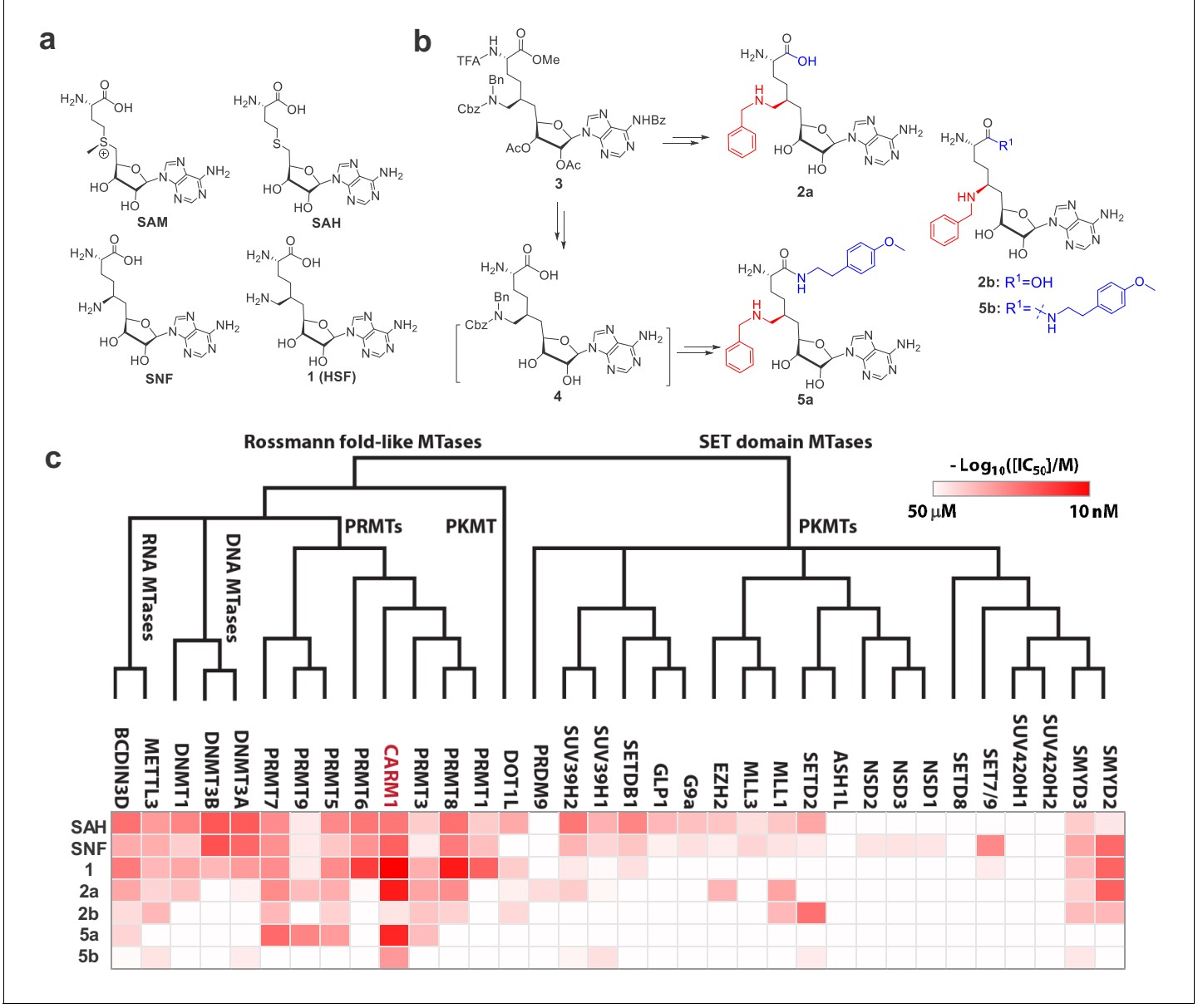

**Figure 1.** Structures, synthesis and target inhibition of SAM analogs. (a) Structures of SAM, SAH, sinefungin (**SNF**) and 6′-homosinefungin (HSF, **1**). (b) Structures and synthetic outline of HSF derivatives **2a** and **5a**, and their structurally related control compounds **2b** and **5b**. (c) IC$_{50}$ heat-map of SAM analogs against 34 methyltransferases. HSF derivatives **2a** and **5a** were identified as potent and selective inhibitors of CARM1; **2b** and **5b** as their respective control compounds.

The online version of this article includes the following figure supplement(s) for figure 1:

**Figure supplement 1.** Synthetic scheme of **1** and **2a** through the precursor **3**.

**Figure supplement 2.** Synthetic scheme of **5a** and **6a** (SKI-73) through the precursor **3**.

**Figure supplement 3.** Synthetic scheme of **5b** and **6b** (SKI-73N).

and **SNF** to engage PRMTs and meanwhile leverages its 6′-methyleneamine group for additional interaction.

To further explore the 6′-region of **HSF**, we synthesized **HSF** derivatives from the same precursor **3** (*Figure 1b*, *Figure 1—figure supplement 2*), by further expanding the 6′-methylene amine moiety with different substituents. The **HSF** derivative **2a** (*Figure 1b*) was identified for its preferential binding to CARM1 with IC$_{50}$ = 30 ± 3 nM and >10 fold selectivity over other seven human PRMTs and 26 methyltransferases of other classes (*Figure 1c*, *Supplementary file 1*-Table A). The structural

difference between **2a** and **1** (*Figure 1b*) suggests that the *N*-benzyl substituent enables **2a** to engage CARM1 through a distinct mechanism (see results below).

With **2a** as a lead, we then explored its α-amino carboxylate moiety with different amides from the common precursor three and then the intermediate **4** (*Figure 1—figure supplement 2*), which led to the discovery of **5a**. This engagement of CARM1 with **2a** is expected to be largely maintained by **5a**. Here, **5a** shows an IC$_{50}$ of 43 ± 7 nM against CARM1 and a >10-fold selectivity over the panel of 33 diverse methyltransferases (*Figure 1c*, *Supplementary file 1*-Table A). In comparison, the negative control compounds **2b** (Bn-SNF) (*Zheng et al., 2012*) and **5b** (*Figure 1b*, *Figure 1—figure supplement 3*), which differ from **2a** and **5a** only by the 6′-methylene group, poorly inhibit CARM1 (IC$_{50}$ = 22 ± 1 μM and 1.91 ± 0.03 μM) (*Figure 1c*, *Supplementary file 1*-Table A). The dramatic increase of the potency of **2a** and **5a** in contrast to **2b** and **5b** supports an essential role of the 6′-methylene moiety upon binding CARM1. Distinguished from the SAM mimics **SAH**, **SNF** and **1** as nonspecific PMT inhibitors, **2a** and **5a** were developed as potent and selective SAM analogs.

## Modes of interaction of 6′-homosinefungin derivatives as CARM1 inhibitors

With **2a** and **5a** characterized as CARM1 inhibitors, we leveraged orthogonal in vitro assays to explore their modes of interaction (*Figure 2a*). To examine whether **2a** and **5a** are SAM- or substrate-competitive, CARM1 inhibition by **2a** and **5a** was assessed in the presence of various concentrations of SAM cofactor and H3 peptide substrate (*Figure 2b,c*). IC$_{50}$ values of **2a** and **5a** showed a linear positive correlation with SAM concentrations, as expected for SAM-competitive inhibitors (*Daigle et al., 2011*; *Luo, 2018*; *Zheng et al., 2012*) .The $K_d$ values of **2a** and **5a** ($K_{d,2a}$ = 17 ± 8 nM; $K_{d,5a}$ = 9 ± 5 nM) were extrapolated from the y-axis intercepts upon fitting the equation IC$_{50}$ = [SAM]×$K_d$/$K_{m,SAM}$+$K_d$ (*Figure 2b*) (*Segel, 1993*). $K_{m,SAM}$ of 0.21 ± 0.09 μM and 0.28 ± 0.14 μM (an averaged $K_{m,SAM}$ = 0.25 μM) for competition with **2a** and **5a**, respectively, can also be derived through the ratio of the y-axis intercepts to the slopes (*Figure 2b* and Materials and methods) (*Segel, 1993*). By contrast, the presence of the H3 peptide substrate had negligible effect on the binding of **2a** and **5a**, indicating their substrate-noncompetitive character (*Figure 2c*). The SAM analogs **2a** and **5a** were thus characterized as SAM-competitive, substrate-noncompetitive inhibitors of CARM1.

For the direct binding of **2a** and **5a** to CARM1, the CARM1-binding kinetics of **2a** and **5a** were examined using surface plasmon resonance (SPR) (*Figure 2d*). The SPR signal progression of **2a** and **5a** fits with a biphasic rather than a mono-phasic binding mode, with the lower $K_{i1,2a}$ = 0.06 ± 0.02 μM, $K_{i1,5a}$ = 0.10 ± 0.01 μM, and the higher $K_{i2,5b}$ = 0.54 ± 0.07 μM, $K_{i2,2a}$ = 0.4 ± 0.1 μM, probably because of the multi-phase binding kinetics of **2a** and **5a** (*Figure 2d*). To cross validate the binding of **2a** and **5a** to CARM1, we conducted an in vitro thermal shift assay, for which ligand binding is expected to increase CARM1's thermal stability (*Blum et al., 2014*). The binding of **2a** and **5a** (at 5 μM concentration) increased the melting temperature ($T_m$) of CARM1 by 4.4°C and 6.5°C, respectively (*Figure 2e*, $T_{m, 2a}$ = 44.2 ± 0.4 °C and $T_{m, 5a}$ = 46.3 ± 0.3 °C versus $T_{m, DMSO}$ = 39.8 ± 0.3 °C as control). By contrast, the binding of SAM and **1** show much reduced effects on $T_m$ of CARM1 (*Figure 2e*, $T_{m, SAM}$ = 40.1 ± 0.3 °C and $T_{m, 1}$ = 42.8 ± 0.4 °C versus $T_{m, DMSO}$ = 39.8 ± 0.3 °C). Therefore, although the affinities of **1**, **2a** and **5a** to CARM1 are comparable (IC$_{50}$ = 13–43 nM, *Figure 1c*), their well-separated effects on $T_m$ suggest that these inhibitors engage CARM1 differentially (see results below). The two orthogonal biochemical assays thus verified the tight binding of **2a** and **5a** with CARM1.

## Structural rationale of 6′-homosinefungin derivatives 5a and 2a as CARM1 inhibitors

To further seek a structural rationale for **5a** and **2a** for CARM1 inhibition, we solved the X-ray structure of CARM1 in complex with **5a** with resolution of 2.00 Å and modeled the CARM1 binding of **2a** (*Figure 3*, Materials and methods). The overall topology of the CARM1–**5a** complex is indistinguishable with the V-shaped subunit of the CARM1 dimer in complex with **SNF** and **1** (details in the next section), which is typical of the Rossmann fold of Class I methyltransferases (*Figure 3a,b*, *Table 1*) (*Luo, 2018*). However, **5a** adopts a noncanonical pose with its 6′-*N*-benzyl moiety in a binding pocket that used to be occupied by the α-amino carboxylate moiety of canonical ligands such as

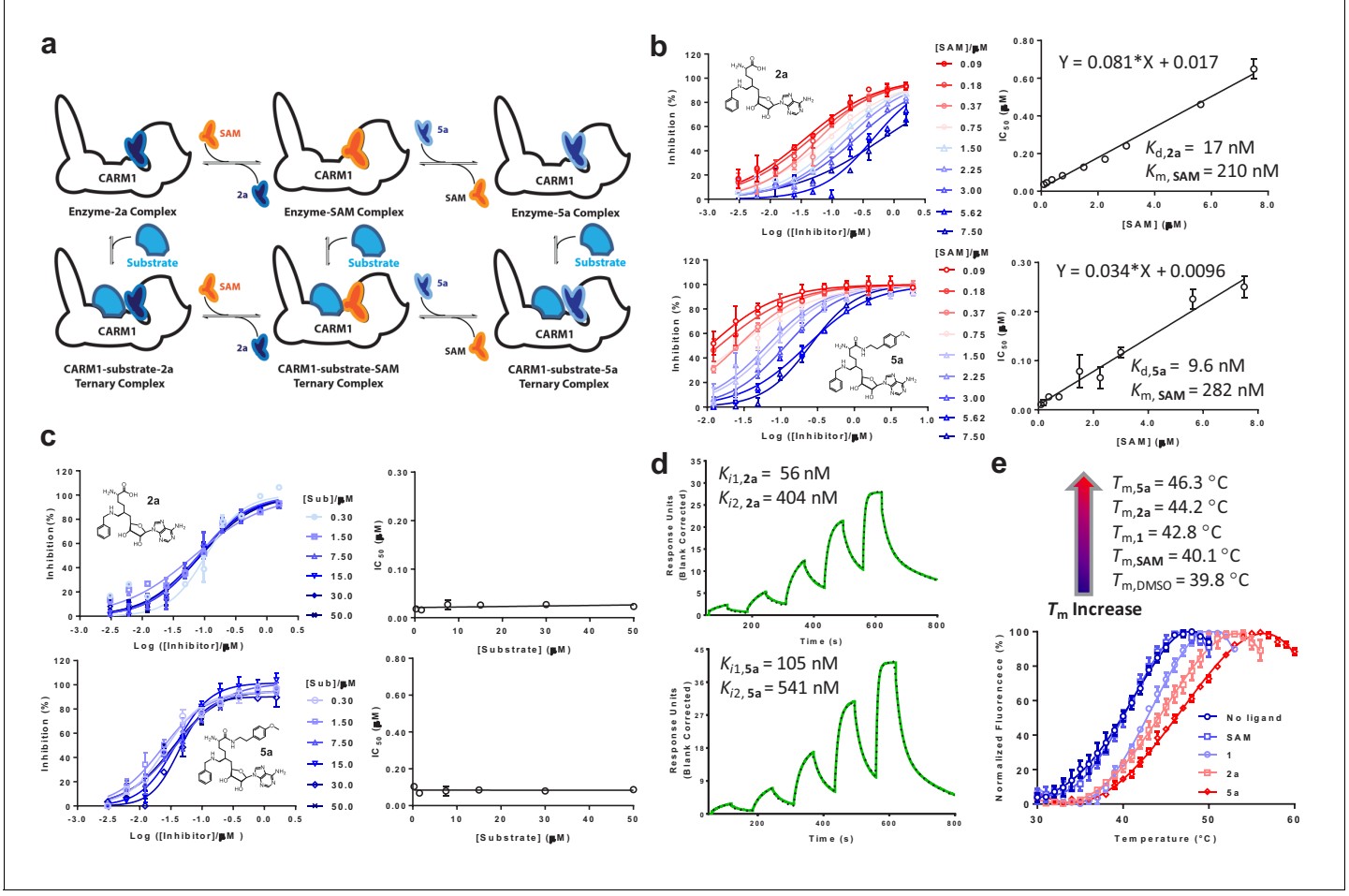

**Figure 2.** In vitro characterization of CARM1 inhibitors 2a and 5a. (a) Schematic description of CARM1 in complex with SAM, **2a** and **5a** in the absence or presence of a substrate peptide. (b, c) $IC_{50}$ of **2a** and **5a** in the presence of varied concentrations of SAM and H3 peptide substrate. $IC_{50}$ data were obtained and presented as the mean of replicates ± standard errors. The $IC_{50}$ values of **2a** and **5a** show a linear increase relative to the SAM concentration but remain near constant as the the substrate concentration increases. Given the SAM competitive character, the $K_d$ values of **2a** and **5a** as well as $K_{d,SAM}$ can be obtained according to $IC_{50} = [SAM] \times K_d / K_{d,SAM} + K_d$. (d) SPR assay for the binding of CARM1 by **2a** and **5a**. Processed sensorgrams upon ligand binding (black dots) were fitted with a biphasic binding model (green line) with $K_{i1,2a}$ = 56 nM (0.06 ± 0.02 μM) and $K_{i2,2a}$ = 404 nM (0.4 ± 0.1 μM); $K_{i1,5a}$ = 105 nM (0.10 ± 0.01 μM) and $K_{i2,5a}$ = 541 nM (0.54 ± 0.07). (e) Thermal shift assay of CARM1 in the absence or presence of SAM, **1**, **2a**, and **5a**. $T_m$ values of 39.8 ± 0.2°C, 40.1 ± 0.5°C, 42.8 ± 0.3°C, 44.2 ± 0.6°C and 46.3 ± 0.3°C (means of triplicates ± standard derivatives) were obtained for apo-CARM1 and CARM1 complexes with 5 μM SAM, **1**, **2a**, and **5a**, respectively.

SAH, SNF and **1** (*Figures 3c* and *4*), while the α-amino methoxyphenethyl amide moiety of **5a** protrudes into the substrate-binding pocket (*Boriack-Sjodin et al., 2016*; *Sack et al., 2011*). This noncanonical mode is consistent with the SAM-competitive character of **5a** (*Figure 2b*). In contrast to the noncanonical mode, Arg168 in the CARM1–**5a** complex adopts an alternative orientation (two possible configurations), accompanied by an altered conformation of Glu257, to accommodate the 6'-*N*-benzyl moiety of **5a** (*Figure 3d*). The α-amino amide moiety of **5a** also engages CARM1 through the combined outcomes of a hydrogen-bond network and hydrophobic interactions with nearby resides (*Figure 3e*). Interestingly, the overlaid structures of CARM1 in complex with **5a** and a substrate peptide implicate a steric clash and thus a potential for binding competition between **5a** and a CARM1 substrate (*Figure 3f*). However, the apparent substrate-noncompetitive character of **5a** (*Figure 2c*) suggests that this steric clash might be avoided if there is no significant energy penalty when the substrate Arg adopts alternative conformation(s).

The binding mode of the CARM1–**2a** complex was modeled via molecular docking followed by molecular dynamics (MD) simulation (Materials and methods). Here, we uncovered two distinct poses of **2a** (Binding Pose 1/2 or **BP1/2**, *Figure 3g*, *Figure 3—figure supplement 1*). **BP1** was

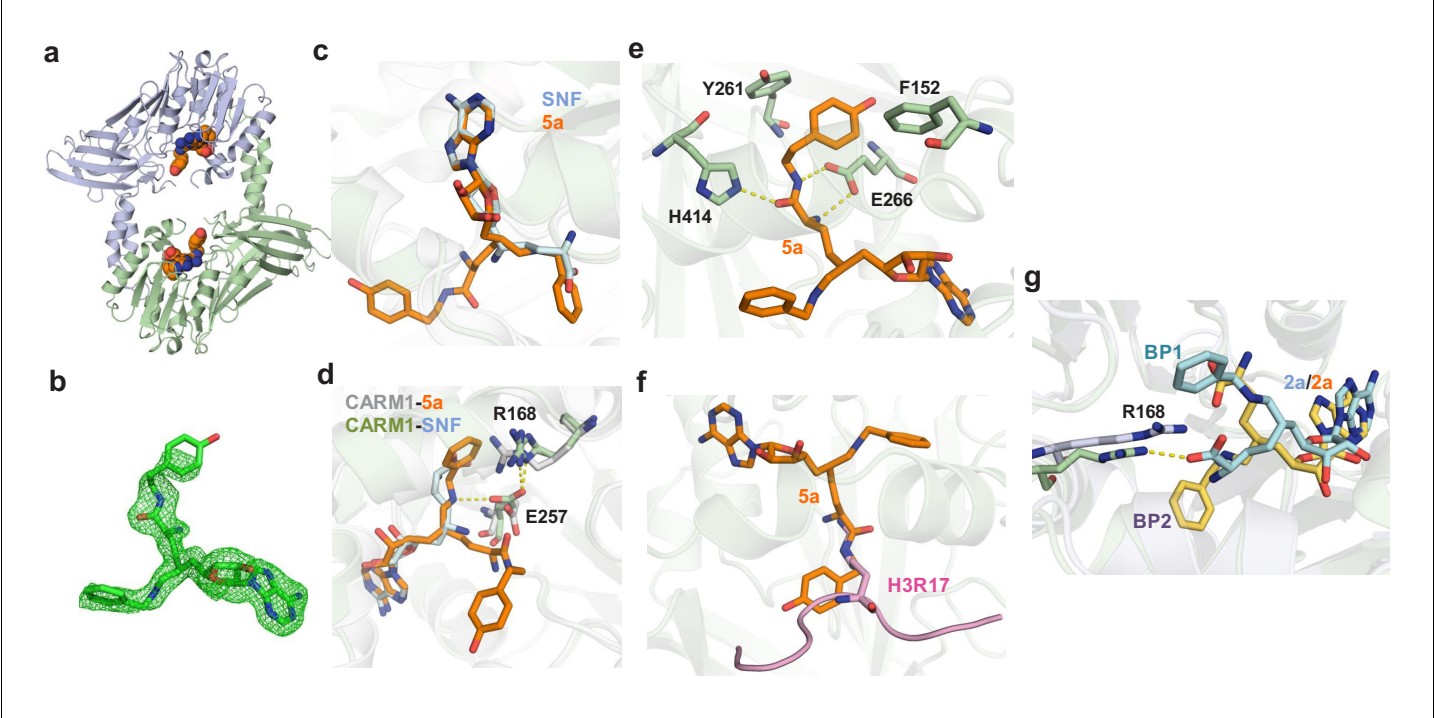

**Figure 3.** Crystal structure or molecular modeling of CARM1 in complex with 5a and 2a. (a) Overview of the Rossmann fold in the X-ray structure of CARM1 with 5a. (b) Total omission electron density map of 5a in the CARM1–5a complex. The total omission electron density map was calculated using SFCHECK, as described in the Materials and methods. The electron density contoured at 1.0 σ is shown for the ligands. (c) Comparison of the binding modes between 5a (noncanonical) and SNF (canonical). The structure of SNF was extracted from a CARM1–SNF–H3R17 complex (PDB 5DX0). (d) Key interactions between CARM1 and ligands in canonical and noncanonical binding modes. The differentiated interactions are highlighted in gray (CARM1) and blue (SNF) for the canonical mode; and in green (CARM1) and orange (5a) for the noncanonical mode. (e) Additional interactions in which the α-amino amide moiety of 5a forms hydrogen bonds with Glu266 and His414 and hydrophobic interactions with Phe152 and Tyr261. (f) Steric clash between the α-amino amide moiety of 5a and an Arg substrate. The structure of the Arg substrate was extracted from a CARM1–SNF–H3R17 complex (PDB 5DX0). (g) Two modeled binding poses (BP1 and BP2) of 2a upon binding CARM1 with the Cβ-Cγ-Cδ-Nε dihedral angle χ3 = 180°; the C4'-C5'-C6'-C7' dihedral angle of −50° for BP1 versus χ3 = −65° C4'-C5'-C6'-C7' dihedral −170° for BP2.

The online version of this article includes the following figure supplement(s) for figure 3:

**Figure supplement 1.** Conformational dynamics of CARM1–2a and CARM1–SNF complexes.

characterized by the direct interaction between the α-amino carboxylate moiety of **2a** and the guanidinium of Arg168, whereas **BP2** features a titled orientation of Arg168 to accommodate the 6'-*N*-benzyl moiety of **2a** (*Figure 3g*, *Figure 3—figure supplement 1*). The BP1 and BP2 of **2a** closely resemble those of **1** and **5a**, respectively, in terms of the orientations of Arg168 and the α-amino carboxylate moiety of the ligands. When the same modeling protocol was applied to the CARM1–SNF complex, only the canonical pose was identified (Materials and methods). Energy calculation indicated that both **BP1** and **BP2** are stable with comparable binding free energies. Interestingly, the side chain configurations of His414 in both **BP1** and **BP2** are different from those in the CARM1–5a complex and the CARM1–SNF complex (*Figure 3g*). Collectively, **5a** and **2a**, though structurally related to the SAM analogs **1** and **SNF**, engage CARM1 via distinct modes of interaction.

Upon comparing the CARM1 structure in complex with **5a** and **2a**, we observed the additional hydrogen-bond and hydrophobic interactions of **5a** that involve its α-amino amide moiety (*Figure 3e*). Interestingly, these interactions do not increase but rather decrease the affinity of **5a** to CARM1 by two-fold ($K_{d,2a}$ = 17 ± 8 nM versus $K_{d,5a}$ = 9 ± 5 nM, *Figure 2b*). By contrast, there is a significant 10-fold increase of affinity between **2b** and **5b** (*Figure 1c*, *Supplementary file 1*-Table A). These observations suggest that, although **5b** facilitates CARM1's engagement better than **2b** via the former's α-amino amide moiety, such an effect is dispensed with in the presence of the 6'-methylene (*N*-benzyl)amine moiety of **5a** and **2a**.

**Table 1.** Crystallography data and refinement statistics of the X-ray structures of CARM1 in complex with **5a**.

| Ligands | 5a |
|---|---|
| PDB Code | 6D2L |
| Data collection | |
| Wavelength (Å) | 0.97918 |
| Space group | $P2_1$ |
| Cell dimensions | |
| a, b, c (Å) | 75.6, 155.6, 95.3 |
| α, β, γ (°) | 90.0, 101.0, 90.0 |
| Resolution (Å) | 50.0–2.00 |
| Unique reflections | 142, 452 |
| Redundancy | 4.5 |
| Completeness (%) | 97.0 |
| I/σ(I) | 10.4 |
| $R_{sym}^a$ | 0.155 |
| $R_{pim}$ | 0.081 |
| Refinement | |
| No. protein molecules/ASU | 6 |
| Resolution (Å) | 50.0–2.00 |
| Reflections used or used/free | 139,748/1400 |
| Rwork(%) | 18.7 |
| Rfree(%) | 23.6 |
| Average B value (Å$^2$) | 30.8 |
| *Protein* | 30.5 |
| *Compound* | 34.8 |
| *Other* | n/a |
| *Water* | 35.5 |
| Number of Atoms | 16,868 |
| *Protein* | 15,819 |
| *Compound* | 253 |
| *Other* | n/a |
| *Water* | 766 |
| RMS bonds (Å) | 0.008 |
| RMS angles (°) | 1.273 |
| Wilson B (Å$^2$) | 29.6 |
| Ramachandran plot | |
| Most favored (%) | 97.1 |
| Additional allowed (%) | 2.9 |
| Outliers (%) | 0.0 |

## The X-ray structure of the CARM1–1(HSF) complex

Given the tight CARM1 binding by **1**, we solved the X-ray structure of CARM1 in complex with **1** (**HSF**) with a resolution of 2.00 Å (*Figure 4*). The overall folding of the CARM1–**1** complex (PDB: 4IKP) is similar to those of **1** in complex with **SNF** and **SAH** (PDB: 2Y1X, 2Y1W) with a V-shape sub-unit in a dimer of dimers (*Figure 4a*, *Table 2*) (*Sack et al., 2011*). However, the CARM1–**1** complex is distinct for multiple configurations of its ligand and interactions via its 6′-methyleneamine moiety

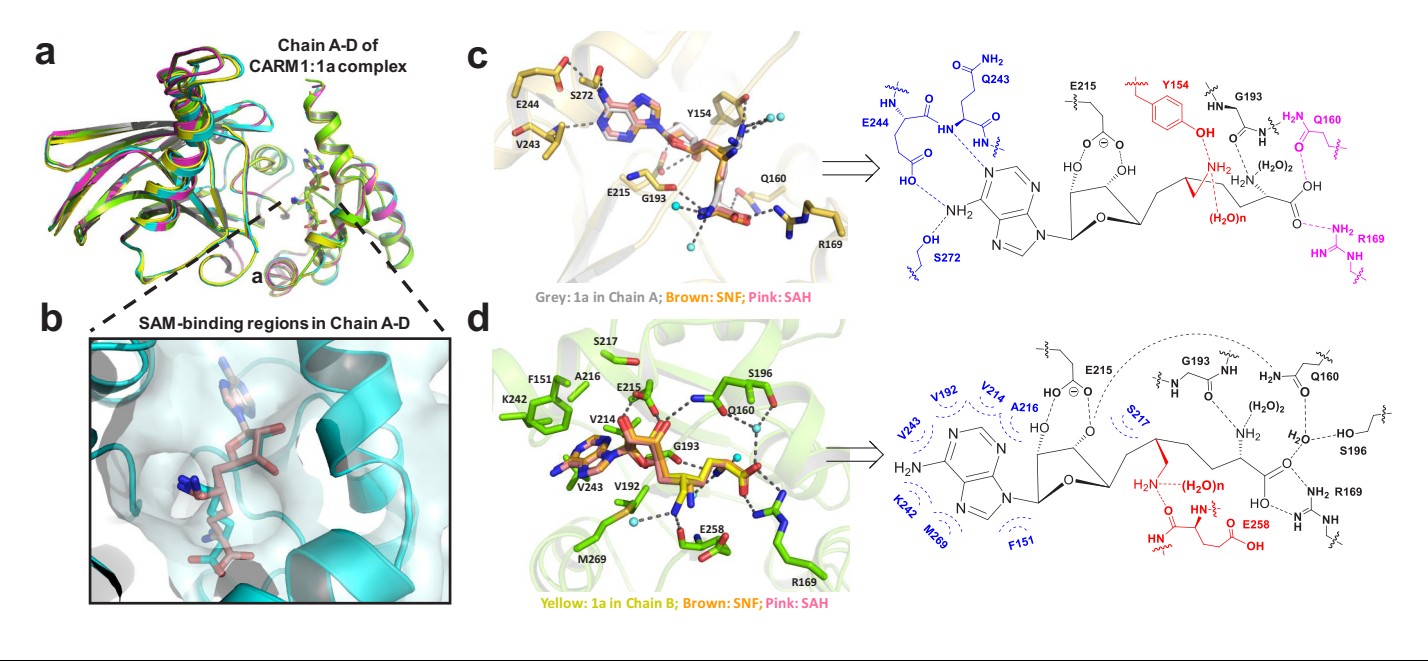

**Figure 4.** Structure of CARM1 in complex with **1** (HSF) in multiple configurations (PDB: 4IKP). (**a**) Overall structures of the CARM1–**1** complex featuring V-shape subunits in a tetramer (Chain A–D). (**b**) Multiple configurations of **1** upon occupying the SAM-cofactor-binding site of CARM1. (**c**) Representative interaction network of **1** (Configuration III, IV in Chain A) upon binding human CARM1, and its comparison with SNF and SAH. Here, we highlight the conserved hydrogen bonds with adenine ring (blue), the 2′,3′-ribosyl hydroxyl/α-amino groups (black), and the distinct interaction network in Chain A for carboxylic moieties/6′-methyleneamine (pink and red). (**d**) Representative interaction network of **1** (Configuration I in Chain B) upon binding human CARM1, and its comparison with SNF and SAH. Here, we highlight the conserved hydrophobic interactions with adenine ring (blue), the conserved hydrogen bond interactions with 2′,3′-ribosyl hydroxyl (black), the α-amino, carboxylic moieties and the distinct interaction network of 6′-methyleneamine (red). The images of SNF and SAH were generated on the basis of PDB files 2Y1W and 2Y1X.

The online version of this article includes the following figure supplement(s) for figure 4:

**Figure supplement 1.** Total omission electron density map of 6′-homosinefungin (HSF, **1**) in the CARM1–**1** complex.

(*Figure 4b–d*, *Figure 4—figure supplement 1*). In the CARM1–**1** complex, **1** can adopt four alternative configurations (Configuration I in Chains B, D; Configuration II in Chain C; Configuration III and IV in Chain A) accompanied with the structural accommodation of the adjacent residues and water hydrogen bonds (*Figure 4b–d*, *Supplementary file 1*-Table B–D). By contrast, only a single configuration of the ligands was observed in **SAH**- or **SNF**-bound CARM1 (*Figure 4c,d*, PDB: 2Y1X, 2Y1W) (*Sack et al., 2011*).

Detailed structural comparison of CARM1 in complex with **SNF**, **SAH** and **1** further revealed that **1** maintains common interactions observed in the CARM1–**SNF** and CARM1–**SAH** complexes with several exceptions (*Figure 4c,d*, *Supplementary file 1*-Table B–D). Most noticeably, the CARM1–**1** complex gains the strong hydrogen bonds via the 6′-methyleneamine moiety of the ligand with (i) the backbone carbonyl of CARM1's Glu258 (Configuration I, II in Chains B, C, D and Configuration III in Chain A) or (ii) the side chain of Tyr154 (Configuration IV in Chain A), together with several less conserved water hydrogen bonds (*Figure 4c,d*, *Supplementary file 1*-Table B–D). By contrast, the 6′-amine of **SNF** in the CARM1–**SNF** complex forms weaker hydrogen bonds with Glu258 and may fewer water hydrogen bonds (*Figure 4c*, *Supplementary file 1*-Table B, C, PDB: 2Y1W). Comparable interactions are completely absent from the CARM1–**SAH** complex (*Figure 4d*, *Supplementary file 1*-Table B–D, PDB: 2Y1X). The desired hydrogen-bond networks of the 6′-methyleneamine moiety of **1** with CARM1, which are present in the CARM1-**1** complex but absent from the CARM1–**SNF** and CARM1–**SAH** complexes, can rationalize the significant decrease of IC$_{50}$ from **SNF** and **SAH** to **1**.

Another key difference among CARM1–**1**, CARM1–**SNF** and CARM1–**SAH** complexes lies in the region around the carboxylate moiety of these ligands. In Chains A and C of the CARM1–**1** complex, the carboxylate moiety of the ligand forms an ionic bond with Arg169 and a hydrogen bond with

**Table 2.** Crystallography data and refinement statistics for the X-ray structures of CARM1 in complex with **1**.

| Ligands | 1 |
| --- | --- |
| PDB code | 4IKP |
| Data collection | |
| Wavelength (Å) | 1.03321 |
| Space group | $P2_12_12_1$ |
| Cell dimensions | |
| *a, b, c* (Å) | 75.1, 98.8, 206.6 |
| α, β, γ (°) | 90.0, 90.0, 90.0 |
| Resolution (Å) | 50.0–2.00 |
| Unique reflections | 104, 330 |
| Redundancy | 8.1 |
| Completeness (%) | 99.8 |
| I/σ(I) | 30.4 |
| $R_{sym}^a$ | 0.086 |
| $R_{pim}$ | 0.032 |
| Refinement | |
| No. protein molecules/ASU | 4 |
| Resolution (Å) | 48.1–2.00 |
| Reflections used or used/free | 103,958 |
| Rwork(%) | 20.3 |
| Rfree(%) | 23.1 |
| Average B value (Å$^2$) | 33.9 |
| *Protein* | 33.4 |
| *Compound* | 29.4 |
| *Other* | n/a |
| *Water* | 42.1 |
| Number of atoms | 11,635 |
| *Protein* | 10,770 |
| *Compound* | 117 |
| *Other* | n/a |
| *Water* | 748 |
| RMS bonds (Å) | 0.007 |
| RMS angles (°) | 1.127 |
| Wilson B (Å$^2$) | 33.9 |
| Ramachandran plot | |
| Most favored (%) | 96.9 |
| Additional allowed (%) | 3.03 |
| Outliers (%) | 0.07 |

Gln160 (*Figure 4b,c,d*, *Supplementary file 1*-Table B, C). Such interactions are absent from CARM1–**SNF** and CARM1–**SAH** complexes (*Figure 4d*). By contrast, in Chains B and D of the CARM1–**1** complex, the same carboxylate moiety forms the ionic bonds with Arg169 and a water hydrogen bond (*Figure 4b,c,d*, *Supplementary file 1*-Table B, C). To accommodate the latter conformation, the Gln160 residue flips toward the 3′-ribosyl hydroxyl moiety of **1** to form a new hydrogen bond (*Figure 4b,c*, *Supplementary file 1*-Table B, C). Similar interaction patterns can also be

found in the CARM1–**SNF** and CARM1–**SAH** complexes (*Figure 4*, *Supplementary file 1*-Table B–D).

With regards to the rest of the CARM1–ligand interactions, the CARM1 complexes with **1**, **SAH** and **SNF** are nearly identical except for slightly altered water hydrogen bonds (*Figure 4*, *Supplementary file 1*-Table S2–S4). Here, the α-amino moiety of these ligands forms hydrogen bonds with the carbonyl backbone of Gly193, as well as two water hydrogen bonds; their 2′,3′-ribosyl hydroxyl groups form two hydrogen bonds with the side chain of CARM1's Glu215; adenine's N1 and N6 form the hydrogen bonds with Asn243 and Glu244/Ser272, respectively; and the adenine ring of these ligands is buried within a hydrophobic pocket. By contrast, there are fewer conserved water hydrogen bonds, such as those involved with the carboxylate and 3′-ribosyl hydroxyl moieties of **1** in Chain A of the CARM1–**1** complex (*Figure 4*, *Supplementary file 1*-Table C). By contrast, adenine-N7 in the CARM1–**SNF** and CARM1–**SAH** complexes forms water hydrogen bonds bridged to Ser272, which are absent from the CARM1–**1** complex (*Figure 4*, *Supplementary file 1*-Table C, D). Collectively, the general high affinity of **1**, **SNF** and **SAH** (*Figure 4*, *Supplementary file 1*-Table B–D) arises from the combined hydrophilic and hydrophobic interactions of these ligands with CARM1. However, in comparison with **SNF** and **SAH**, **1** gains the extra interactions via its 6′-methyleneamine moiety (*Figure 4c,d*, *Supplementary file 1*-Table B–D). In addition, **1** adopts the canonical pose with its α-amino carboxylate moiety interacting with Arg168, which is similar to that of **SNF** and **SAH** but different from the noncanonical pose of **2a** and **5a**, upon binding CARM1 (*Figure 4d*).

## A pro-drug-like 6′-homosinefungin derivative as a cell-active CARM1 inhibitor

Although the in vitro characterization demonstrated the potency and selectivity of **2a** and **5a** against CARM1, we anticipated their poor membrane permeability as observed for structurally related analogs such as **SAH** and **SNF** (*Figure 1a*) (*Boriack-Sjodin et al., 2016*; *Sack et al., 2011*). The lack of membrane penetration is probably due to their primary amine moiety, which has pKa of ~10 and is fully protonated at a physiological pH of 7.4. Given the essential roles of the 9′−amine moiety of **2a** and **5a** in CARM1 binding (*Figure 3e*), we envisioned overcoming the membrane permeability issue through a pro-drug strategy by cloaking this amine moiety with a redox-triggered trimethyl-locked quinone butanoate moiety (**TML**, *Figure 5a*) (*Levine and Raines, 2012*). We thus prepared **6a** and its control compound **6b** by derivatizing **5a** and **5b** with the **TML** moiety (*Figure 1b*, *Figure 1—figure supplements 1* and *2*). To assess the cellular activity of **6a**, we relied on our prior knowledge that CARM1 methylates the Arg1064 of BAF155, a core component of the SWI/SNF chromatin-remodeling complex, and *CARM1* knockout abolishes this posttranslational modification in MCF-7 cells (*Wang et al., 2014a*). Treatment of MCF-7 cells with 10 µM of **6a** fully suppressed this methylation mark, whereas treatment with **2a** and **5a** did not affect this mark (*Figure 5b*). We thus demonstrated the prodrug-like cellular activity of **6a**.

## Characterization of 6a (SKI-73) as a chemical probe of CARM1

To further evaluate **6a** as a chemical probe against CARM1, we quantified the efficiency by which **6a** engages CARM1 in a cellular context and thus suppresses the CARM1-dependent invasion by breast cancer cells. Because of the pro-drug character of **6a** and its control compound **6b**, we first developed quantitative LC-MS/MS methods to examine their cellular fates for CARM1 engagement (see Materials and methods). Upon treatment of MDA-MB-231 cells with **6a**, we observed its time- and dose-dependent intracellular accumulation (*Figure 5c*, *Figure 5—figure supplements 1* and *2*). We anticipated the conversion of the pro-drug **6a** into **5a**, but a striking finding is that **6a** can also be readily processed into **2a** inside cells (*Figure 5c*, *Figure 5—figure supplements 1* and *2*). Remarkably, >100 µM of **2a** can be accumulated inside cells for 2 days after 6 hr treatment with a single dose of 5–10 µM **6a**. This observation probably reflects a slow efflux and thus effective intracellular retention of **2a** due to its polar α-amino acid zwitterion moiety. Given that cellular CARM1 inhibition is involved with multiple species (**2a**, **5a** and **6a**) in competition with SAM, we modeled the ligand occupancy of cellular CARM1 on the basis of their $K_d$ values ($K_{d,2a}$=17 nM, $K_{d,5a}$=9 nM, $K_{d,6a}$=0.28 µM and $K_{d,SAM} \approx K_{m,SAM}$=0.25 µM) and MS-quantified intracellular concentrations (*Figure 5d*, *Figure 5—figure supplement 3*, *Equations 5–7*, see Materials and methods). The SAM cofactor, whose intracellular concentration was determined to be 89 ± 16 µM (*Figure 5c,d*), is expected to

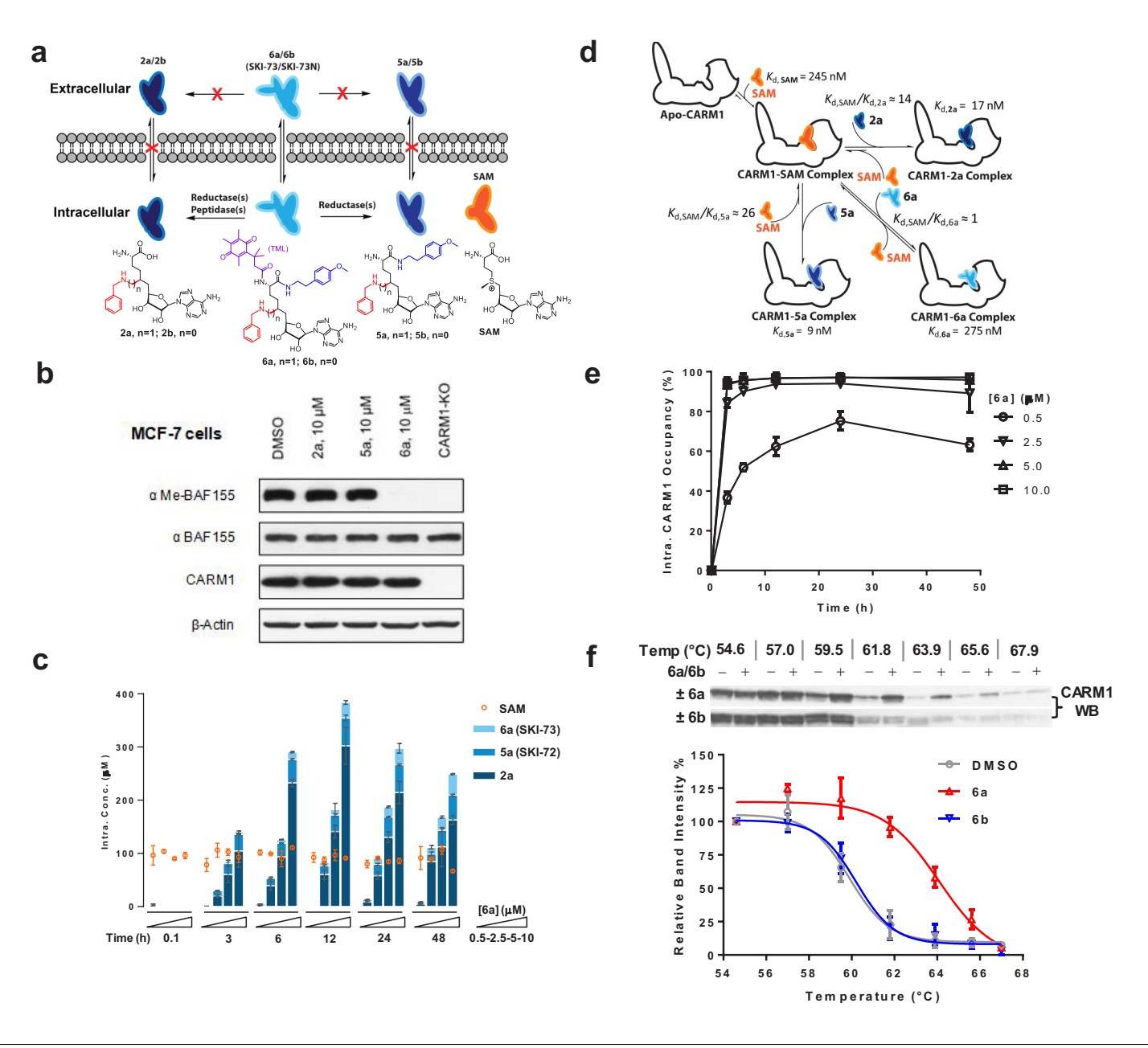

**Figure 5.** Characterization of cellular activity of **6a** as a chemical probe. (a) Schematic description of the extracellular and intracellular fates of **2a**, **5a** and **6a**. Extracellularly, **2a**, **5a** and **6a** are stable; only **6a** can readily penetrate cell membrane. Intracellularly, **6a** can be processed into **5a** and **2a**. Given the poor membrane permeability of **2a** and **5a**, they are accumulated within cells at high concentrations. (b) CARM1 inhibition of **2a**, **5a** and **6a** in MCF-7 cells with BAF155 methylation as a mark. MCF-7 cells were treated with 10 μM of **2a**, **5a** and **6a** for 48 hr. The ratios between me-BAF155 and BAF155 were quantified as a cellular reporter of CARM1 inhibition. DMSO-treated MCF-7 cells and MCF-7 *CARM1*-KO cells were used as negative and positive controls, respectively. (c) MS-based quantification of intracellular concentrations of **2a**, **5a**, **6a** and SAM. These compounds were accumulated within MDA-MB-231 cells in a dose- and time-dependent manner. In comparison, the intracellular concentration of SAM remains a constant at 89 ± 16 μM. (d) Schematic description of intracellular engagement of CARM1 by **6a**, **5a** and **2a** in the presence of the SAM cofactor (*Equations 5–7*). (e) Modeled ligand occupancy of CARM1 with **2a**, **5a** and **6a** as ligands in competition with the SAM cofactor. Percentage of competitive CARM1 occupancy was calculated on the basis of the concentrations of ligands (SAM, **2a**, **5a** and **6a**, *Figure 5c*) and their $K_d$ values (*Figure 2c*). (f) Cellular thermal shift assay (CETSA) of CARM1. Representative western blots of CARM1 in MDA-MB-231 cells upon the treatment of 15 μM **6a** or its negative control **6b** for 48 hr with DMSO treatment as reference. The relative intensity of CARM1 was quantified. The $T_m$ values were determined at the 50% loss of the relative intensity signals with $T_{m,6a}$ = 63.9 ± 0.3 °C, $T_{m,6b}$ = 60.2 ± 0.6 °C and $T_{m,DMSO}$ = 59.6 ± 0.2 °C.

The online version of this article includes the following figure supplement(s) for figure 5:

*Figure 5 continued on next page*

*Figure 5 continued*

**Figure supplement 1.** LC-MS/MS working curves for quantification of the analytes **6a** (SKI-73), **5a** and **2a**, with **6b** (SKI-73N), **5b**, and **2b** as internal standards.

**Figure supplement 2.** LC-MS/MS working curves for quantification of the analytes **6b** (SKI-73N), **5b** and **2b**, with **6a** (SKI-73), **5a**, and **2a** as internal standards.

**Figure supplement 3.** IC$_{50}$ of **6a** (SKI-73) against CARM1.

**Figure supplement 4.** Modeled ligand occupancy of SMYD2 by **2a**.

**Figure supplement 5.** Microsome stability of **6a** (SKI-73).

**Figure supplement 6.** Cellular uptake and intracellular fate of **6a** (SKI-73) and **6b** (SKI-73N).

occupy >99.5% CARM1 with residual <0.5% as the apo-enzyme under a native setting. With single doses of **6a** of 2.5–10 μM, the combined CARM1 occupancy by **2a**, **5a** and their pro-drug precursor **6a** rapidly reached the plateau of >95% within 6 hr, and was maintained at this level for at least 48 hr (*Figure 5e*, *Figure 5—figure supplements 1* and *2*). Notably, treatment with **6a** concentrations as low as 0.5 μM is sufficient to reach 60% target engagement within 10 hr and to maintain this occupancy for 48 hr (*Figure 5e*, *Figure 5—figure supplements 1* and *2*). The time- and dose-dependent progression of the CARM1 occupancy by these ligands thus provides quantitative guidance upon the treatment of MDA-MB-231 cells with **6a**.

Given that **2a** is the predominant metabolic product of **6a** within cells (*Figure 5c,e*) and also shows certain affinity to SMYD2 (~10 fold higher IC$_{50}$ in comparison with CARM1, *Figure 1c*, *Supplementary file 1*-Table A), we evaluated SMYD2 engagement of **2a** for its potential off-target effect. In a similar manner to that described for the ligand occupancy of cellular CARM1, we modeled the occupancy of cellular SMYD2 by **2a** on the basis of $K_{d,2a}$=150 nM and $K_{d,SAM}$=60 nM for SMYD2 (*Figure 5e*, *Figure 5—figure supplement 4*, Materials and methods). Largely because of the high affinity of **2a** to SAM and thus a 37-fold larger $K_{d,2a}/K_{d,SAM}$ ratio of SMYD2 relative to CARM1 ($K_{d,2a}/K_{d,SAM}$ of 2.5 and 0.068 for SMYD2 and CARM1, respectively), the occupancy of SMYD2 by **2a** is below 20% (*Figure 5e*, *Figure 5—figure supplement 4*) under the efficacy doses of **6a** (see cellular data for **6a** below). We were thus less concerned about the SMYD2-associated off-target effect under our assay conditions.

The metabolic stability of **6a** was also evaluated with a microsomal stability assay (*Figure 5c,e*, *Figure 5—figure supplement 5*, see Materials and methods). In the presence of rat liver microsomes, **6a** showed decent stability with 24% residual **6a** after one-hour incubation. Here, the conversion of **6a** into **5a** accounted for 40% of the microsome-processed **6a**; no production of **2a** was detected. Such observation suggests that NQO1, the putative enzyme candidate to reduce the **TML** moiety in **6a** or **6b**, is present in microsomes as well as in tumor cells (*Dias et al., 2018*; *Huang et al., 2016*). By contrast, peptidase enzymes that are expected to process **5a** into **2a** are absent from microsomes but rich in tumor cells.

We then conducted a cellular thermal shift assay (CETSA), in which ligand binding is expected to increase CARM1's thermal stability in a cellular context (*Jafari et al., 2014*). Our data showed that the treatment of MDA-MB-231 cells with **6a** but with not the control compound **6b** increased cellular $T_m$ and thus the thermal stability of CARM1 by 4.3 ± 0.6°C (*Figure 5f*). The distinct effect of **6a** in contrast to **6b** on the cellular $T_m$ of CARM1 aligns well with the 4.1–6.2°C difference in the in vitro $T_m$ of CARM1 upon binding **2a** and **5a** versus SAM (*Figure 2e*). Here, **6b** can penetrate cell membranes and be processed into **5b** and **2b** in a similar manner as **6a** (*Figure 5—figure supplement 6*). These observations thus present the cellular evidence to show that CARM1 engages **2a** and **5a**.

To further characterize **6a** as a CARM1 chemical probe, we examined the Arg1064 methylation of BAF155 and the Arg455/Arg460 methylation of PABP1, two well-characterized cellular methylation marks of CARM1, upon treating MDA-MB-231 cells with **6a** (*Lee and Bedford, 2002*; *Wang et al., 2014a*). These methylation marks can be fully suppressed by **6a** in a dose-dependent manner (*Figure 6a*). The resultant EC$_{50}$ values of 0.45–0.75 μM (*Figure 6b*) are well correlated with the modeled 60% cellular occupancy of CARM1 upon treatment with 0.5 μM **6a** for 48 hr (*Figure 5e*). By contrast, the treatment of the negative control compound **6b** showed no effect on these methylation marks (*Figure 6a*). We therefore demonstrated the robust use of **6a** (**SKI-73**) as a CARM1 chemical probe and of **6b** (**SKI-73N**) as its control compound.

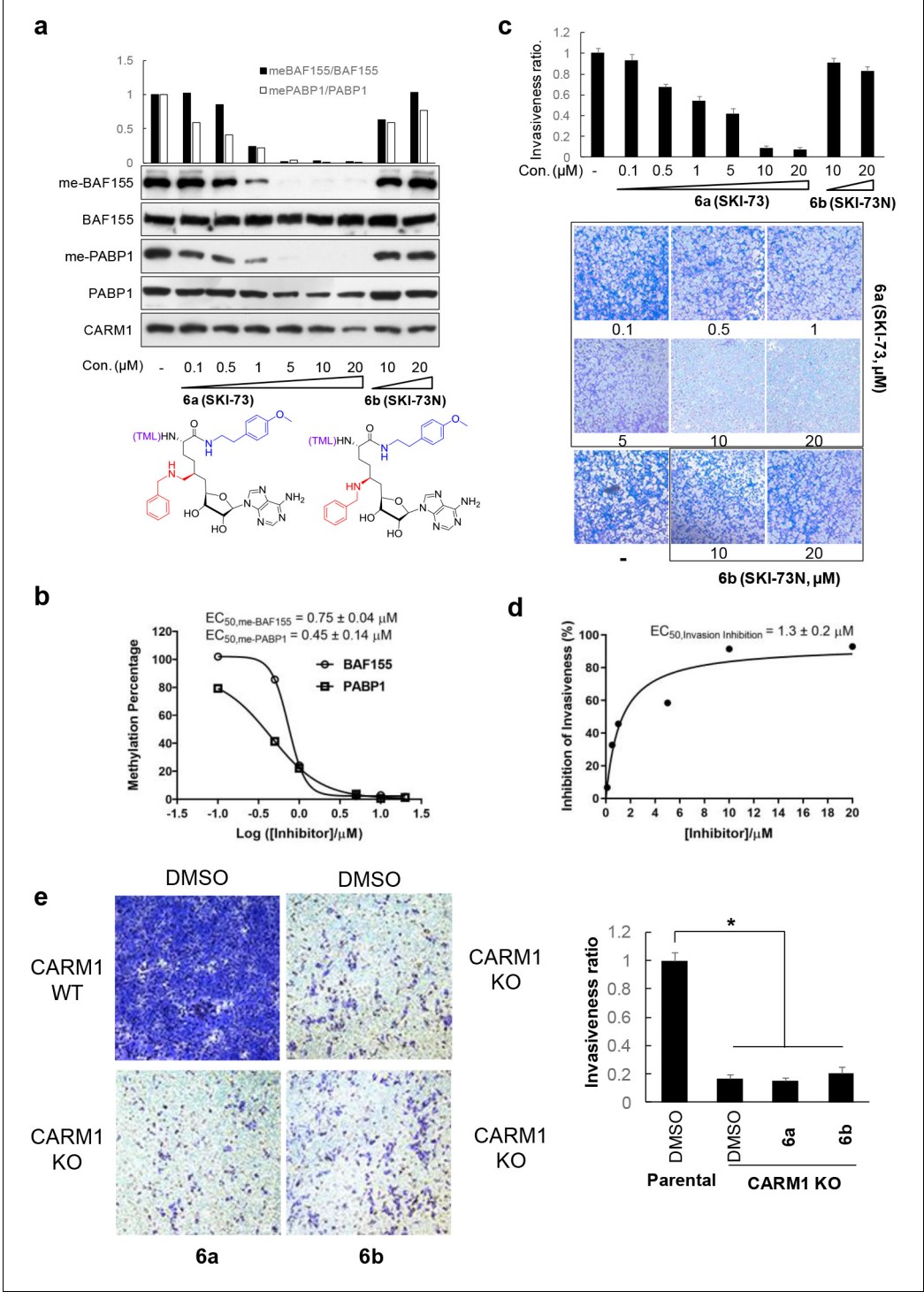

**Figure 6.** Biological outcomes of CARM1 inhibition by **6a** in MDA-MB-231 cells. (**a**) Dose-dependent depletion of BAF155 methylation and PABP1 methylation by 6a. BAF155 methylation and PABP1 methylation, two marks of the CARM1-specific methyltransferase activity, were examined upon the treatment of **6a** and its structural analog **6b** (negative control compound) for 48 hr. Western Blot analysis was then conducted to quantify the relative intensities of the methylated versus total proteins (BAF155 and PABP1, two replicates with a representative one shown). (**b**) $EC_{50}$ of the methylation depletion of BAF155 and PABP1. The relative intensity of the methylated versus total BAF155 or PABP1 was plotted against log[**6a**], with the resultant $EC_{50}$ obtained upon fitting a standard sigmoid curve using GraphPad Prism. (**c**) Inhibition of cell invasion by **6a**. Representative images of the trans-well migration of MDA-MB-231 cells a shown upon treatment with various concentrations of **6a** (**SKI-73**) or its control

*Figure 6 continued on next page*

*Figure 6 continued*

compound **6b** (SKI-73N) for 16 hr. Invasive cells were fixed and stained with crystal violet. The invasiveness ratios were determined using the relative cell invasion of the treatment of **6a** or **6b** versus DMSO treatment. (**d**) $EC_{50}$ of invasion inhibition by **6a**. The invasiveness ratios were plotted as a function of the concentration of **6a**. $EC_{50}$ of $1.3 \pm 0.2$ μM was obtained upon fitting a standard sigmoid curve using GraphPad Prism. (**e**) Effect of **6a** on cell invasion in combination with *CARM1*-KO. Representative images of the trans-well migration of parent and *CARM1*-KO MDA-MB-231 cells are shown upon treatment with DMSO, **6a** or **6b** for 16 hr. The results were analyzed in a similar manner to that described for *Figure 5c,d*. Statistical analysis was carried out to calculate mean ± standard derivation (N = 5) and to perform two-tailed paired t-tests *, p=0.05.

The online version of this article includes the following figure supplement(s) for figure 6:

**Figure supplement 1.** Viability of parental and CARM1-*KO* MDA-MB-231 cells upon treatment with SKI-73 (**6a**) and its control compound SKI-73N (**6b**).

## Inhibition of in vitro invasion but not proliferation of breast cancer cells by SKI-73 (6a)

After demonstrating the utility of **SKI-73** (**6a**) as a chemical probe for CARM1, we examined whether chemical inhibition of CARM1 can recapitulate biological outcomes that are associated with CARM1 knockout (*CARM1*-KO) (*Wang et al., 2014a*) . Our prior work showed that CARM1's methyltransferase activity is required for invasion of MDA-MB-231 cells (*Wang et al., 2014a*). We thus conducted a matrigel invasion assay with MDA-MB-231 cells in the presence of **6a**. Relative to the control treatment with DMSO, treatment with **SKI-73** (**6a**) but not its negative control compound **SKI-73N** (**6b**) suppressed the invasion of MDA-MB-231 cells ($EC_{50}$ = 1.3 μM) (*Figure 6c,d*). The treatment with $\geq 10$ μM **6a** produced the maximal 80% suppression of the invasion by MDA-MB-231 relative to the DMSO control, which is comparable with the phenotype of *CARM1*-KO (*Figure 6e*). Critically, no further inhibition by **6a** on the invasiveness was observed upon **6a** treatment (in comparison with the treatment with DMSO or **6b** treatment) of MDA-MB-231 *CARM1*-KO cells (*Figure 6e*). Notably, treatment with **6a** and **6b** under the current condition has no apparent impact on the proliferation of parental or *CARM1*-KO MDA-MB-231 cells (*Figure 6—figure supplement 1*), consistent with the intact proliferation upon treatment with other CARM1 chemical probes (*Drew et al., 2017*; *Greenblatt et al., 2018*; *Nakayama et al., 2018*). These results suggest that **SKI-73** (**6a**) and *CARM1* knockout perturb the common, proliferation-independent biological process and then suppresses 80% of the invasiveness of MDA-MB-231 cells. We thus characterized **SKI-73** (**6a**) as a chemical probe that can be used to interrogate the CARM1-dependent invasion of breast cancer cells.

## A scRNA-seq and cell-cycle-aware algorithm reveals CARM1-dependent epigenetic plasticity

Because of the advancement of scRNA-seq technology, stunning subpopulation heterogeneity has been uncovered even for well-defined cellular types (*Tanay and Regev, 2017*). In the context of tumor metastasis, including its initial invasion step, epigenetic plasticity is required to allow a small subset of tumor cells to adapt distinct transcriptional cues for neo-properties (*Chatterjee et al., 2018*; *Flavahan et al., 2017*; *Wu et al., 2019*). To explore the feasibility of dissecting the CARM1-dependent, invasion-prone subset of MDA-MB-231 breast cancer cells, we formulated a cell-cycle-aware algorithm of scRNA-seq analysis and dissected those subpopulations that were sensitive to CARM1 perturbation (*Figure 7a*, see Materials and methods). Here we conducted 10 × Genomics droplet-based scRNA-seq of 3232, 3583 and 4099 individual cells (a total of 10,914 cells) exposed to 48 hr treatment with **SKI-73** (**6a**), **SKI-73N** (**6b**) and DMSO, respectively. Guided by Silhouette analysis, cell-cycle-associated transcripts were identified as dominant signatures of subpopulations (*Figure 7—figure supplements 1–18*). These signatures naturally exist for proliferative cells and are not expected to be specific for the invasive phenotype. To dissect the subpopulation-associated transcriptomic signatures of invasive cells, we included one additional layer for hierarchical clustering by first classifying the individual cells into $G_0/G_1$, S, and $G_2/M$ stages (6885, 1520 and 2509 cells, respectively) (*Figure 7—figure supplement 6*, *Supplementary file 1*-Table E), and then conducted the unsupervised clustering within each cell-cycle-aware subset (*Figure 7b*, *Figure 7—figure supplements 19–30*, *Supplementary file 1*-Table E). To resolve the subpopulations without redundant

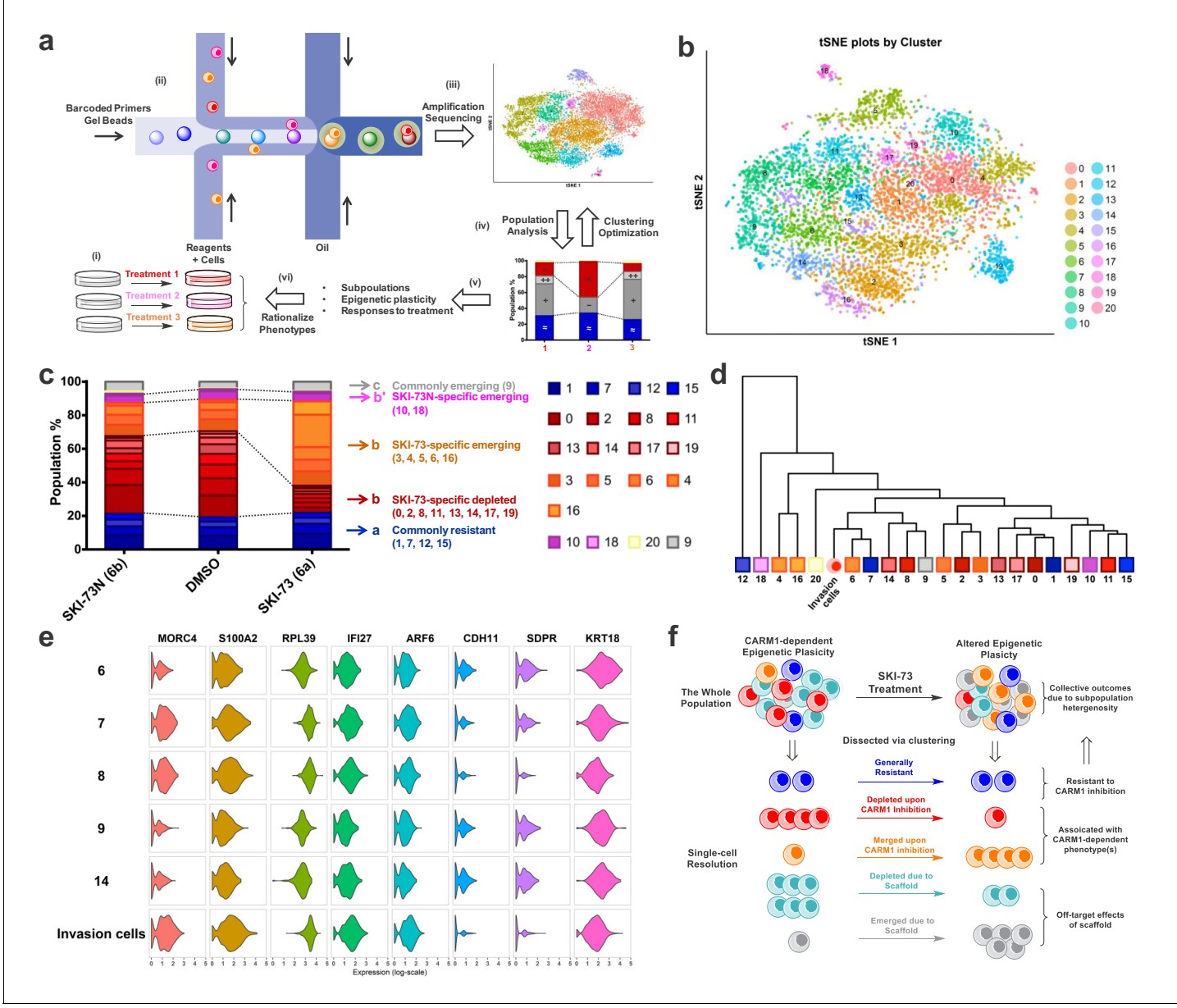

**Figure 7.** scRNA-seq analysis of MDA-MB-231 cells upon CARM1 perturbation. (**a**) Schematic description of scRNA-seq analysis algorithms. (**b**) tSNE plots of the 21 clustered subpopulations of the $G_0/G_1$-phase cells treated with **SKI-73N** (**6b**), DMSO and **SKI-73** (**6a**). (**c**) Population analysis of the 21 clustered subpopulations of the $G_0/G_1$-phase cells upon treatment with **SKI-73N** (**6b**) or **SKI-73** (**6a**). (**d**) Phylogenic tree of the 21 clustered subpopulations of the $G_0/G_1$-phase cells and invasive cells. (**e**) Violin plots of the representative transcripts of $G_0/G_1$-phase invasion cells that distinguish Subpopulation-8 from the closely related Subpopulations 6, 7, 9, and 14 of $G_0/G_1$-phase cells. (**f**) Working modeling of CARM1-dependent epigenetic plasticity perturbed by **SKI-73** (**6a**).

The online version of this article includes the following figure supplement(s) for figure 7:

**Figure supplement 1.** Quality control of the cells that were subjected to scRNA-seq analysis.

**Figure supplement 2.** Silhouette analysis of the combined cell population treated with DMSO, **SKI-73** (**6a**) or **SKI-73N** (**6b**) guided by the resolution granularity.

**Figure supplement 3.** tSNE plot of the subpopulations of the combined cells treated with DMSO, **SKI-73** (**6a**) or **SKI-73N** (**6b**).

**Figure supplement 4.** $G_2/M$- and S-phase scores of the ten clusters within the combined cell population treated with DMSO, **SKI-73** (**6a**) or **SKI-73N** (**6b**).

**Figure supplement 5.** tSNE plot with cell-cycle awareness for the combined cell population treated with DMSO, **SKI-73** (**6a**) or **SKI-73N** (**6b**).

**Figure supplement 6.** Assignment of cell-cycle stages to the cells treated with **SKI-73N** (**6b**), DMSO, or **SKI-73** (**6a**).

**Figure supplement 7.** Silhouette analysis of the DMSO-treated cells guided by the resolution granularity.

*Figure 7 continued on next page*

*Figure 7 continued*

**Figure supplement 8.** tSNE plot of the subpopulations of the DMSO-treated cells.

**Figure supplement 9.** $G_2$/M- and S-phase scores of the 11 clustered subpopulations of DMSO-treated cells.

**Figure supplement 10.** tSNE plot with cell-cycle awareness for the DMSO-treated cells.

**Figure supplement 11.** Silhouette analysis of the **SKI-73N(6b)**-treated cells guided by the resolution granularity.

**Figure supplement 12.** tSNE plot of the subpopulations of **SKI-73N(6b)**-treated cells.

**Figure supplement 13.** $G_2$/M- and S-phase scores of the six clustered subpopulations of **SKI-73N(6b)**-treated cells.

**Figure supplement 14.** tSNE plot with cell-cycle awareness for the **SKI-73N(6b)**-treated cells.

**Figure supplement 15.** Silhouette analysis of the **SKI-73(6a)**-treated cells guided by resolution granularity.

**Figure supplement 16.** tSNE plot of the subpopulations of **SKI-73(6a)**-treated cells.

**Figure supplement 17.** $G_2$/M- and S-phase scores of the four clustered subpopulations within the **SKI-73(6a)**-treated cells.

**Figure supplement 18.** tSNE plot with cell-cycle awareness for the **SKI-73(6a)**-treated cells.

**Figure supplement 19.** Quality control for the $G_0$/$G_1$-phase cells subjected to scRNA-seq analysis.

**Figure supplement 20.** Silhouette analysis, entropy analysis and Fisher's exact test for the classified $G_0$/$G_1$-phase cells after the treatment with **SKI-73N** (**6b**), DMSO or **SKI-73** (**6a**).

**Figure supplement 21.** tSNE plot of the subpopulations of the classified $G_0$/$G_1$-phase cells after treatment with **SKI-73N** (**6b**), DMSO or **SKI-73** (**6a**).

**Figure supplement 22.** tSNE plot of the classified $G_0$/$G_1$-phase cells with awareness of their three treatment origins: **SKI-73N** (**6b**), DMSO or **SKI-73** (**6a**).

**Figure supplement 23.** Quality control of the S-phase cells that were subjected to scRNA-seq analysis.

**Figure supplement 24.** Silhouette analysis, entropy analysis and Fisher's exact test of the classified S-phase cells after treatment with **SKI-73N** (**6b**), DMSO or **SKI-73** (**6a**).

**Figure supplement 25.** tSNE plot of the subpopulations of the classified S-phase cells after treatment with **SKI-73N** (**6b**), DMSO or **SKI-73** (**6a**).

**Figure supplement 26.** tSNE plot of the classified S-phase cells with awareness of their three treatment origins: **SKI-73N** (**6b**), DMSO and **SKI-73** (**6a**).

**Figure supplement 27.** Quality control for the $G_2$/M-phase cells that were subjected to scRNA-seq analysis.

**Figure supplement 28.** Silhouette analysis, entropy analysis and Fisher's exact test of the classified $G_2$/M-phase cells after treatment with **SKI-73N** (**6b**), DMSO or **SKI-73** (**6a**).

**Figure supplement 29.** tSNE plot of the subpopulations of the classified $G_2$/M-phase cells after the treatment with **SKI-73N** (**6b**), DMSO or **SKI-73** (**6a**).

**Figure supplement 30.** tSNE plot of the classified $G_2$/M-phase cells with awareness of their three treatment origins: **SKI-73N** (**6b**), DMSO and **SKI-73** (**6a**).

**Figure supplement 31.** Population analysis for the S-phase cells after treatment with **SKI-73** (**6a**) or **SKI-73N** (**6b**).

**Figure supplement 32.** Population analysis of the $G_2$/M-phase cells after treatment with **SKI-73** (**6a**) or **SKI-73N** (**6b**).

**Figure supplement 33.** tSNE plot of the total cell population with five origins: **SKI-73N(6b)**-, DMSO- and **SKI-73(6a)**-treated cells, invasive cells and CARM1-*KO* cells.

**Figure supplement 34.** Silhouette analysis of invasive cells guided by the resolution granularity.

**Figure supplement 35.** tSNE plot of the subpopulations of invasion cells.

**Figure supplement 36.** $G_2$/M- and S-phase scores of the 10 clustered subpopulation of invasive cells.

**Figure supplement 37.** tSNE plot (center) with cell-cycle awareness and cell-cycle assignment (right upper corner) of invasive cells.

**Figure supplement 38.** tSNE plot of the total $G_0$/$G_1$-phase cells analyzed by scRNA-seq: **SKI-73N(6b)**-, DMSO- and **SKI-73(6a)**-treated cells and invasive cells.

**Figure supplement 39.** tSNE plot of all of the S-phase cells analyzed by scRNA-seq: **SKI-73N(6b)**-, DMSO- and **SKI-73(6a)**-treated cells and invasive cells.

**Figure supplement 40.** tSNE plot of the total $G_2$/M-phase cells analyzed by scRNA-seq: **SKI-73N(6b)**-, DMSO- and **SKI-73(6a)**-treated cells and invasive cells.

**Figure supplement 41.** Unsupervised correlation analysis of the S-phase subpopulations.

**Figure supplement 42.** Unsupervised correlation analysis of the $G_2$/M-phase subpopulations.

**Figure supplement 43.** Heatmap of representative cancer-associated genes for comparison between invasive cells and the 21 subpopulations of $G_0$/$G_1$ cells.

---

clustering, we developed an entropy analysis method and relied on the Fisher Exact test (see Materials and methods). The optimal scores of the combined methods were implemented to determine the numbers of cluster for each subset (*Figure 7b*, *Figure 7—figure supplements 20*, *24* and *28*) (*Butler et al., 2018*). The cell-cycle-aware algorithm allowed the clustering of these cells according to the three cell cycle stages under the three treatment conditions and resulted in 21, 7 and 6 subpopulations in $G_0$/$G_1$, S, and $G_2$/M phases, respectively (*Figure 7b*, *Figure 7—figure supplements 21*, *25* and *29*, *Supplementary file 1*-Tables F–H). Notably, the 48 hr treatments with **SKI-73** (**6a**) or **SKI-73N** (**6b**) had no effect on the cell cycle, as indicated by their comparable cell-cycle distribution patterns (*Figure 7—figure supplement 6*; *Supplementary file 1*-Table E), a finding that is consistent with intact proliferation after all three treatments (*Figure 6—figure supplement 1*).

## CARM1-associated epigenetic plasticity of breast cancer cells at single-cell resolution

With the 21, 7 and 6 subpopulations clustered into the $G_0/G_1$, S, and $G_2$/M stages, respectively, we then conducted population analysis, comparing SKI-73 (6a) and SKI-73N (6b) against DMSO (*Figure 7c*, *Figure 7—figure supplements 31* and *32*, *Supplementary file 1*-Table F–H). These subpopulations can be readily classified into five distinct categories according to their cell-cycle-aware responses to SKI-73 (6a) and SKI-73N (6b) treatment: commonly resistant/emerging/depleted versus differentially depleted/emerging (SKI-73/SKI-73N- or 6a/6b-specific) (*Figure 7c*, *Figure 7—figure supplements 31* and *32*, *Supplementary file 1*-Table F–H). Here, we are particularly interested in the SKI-73(6a)-specific depleted subpopulations (Subpopulation 0/2/8/11/13/14/17/19 of $G_0/G_1$-phase cells and 3 of S-phase cells) as the potential invasion-associated subpopulations, given their sensitivity to SKI-73 (6a) but not its control compound SKI-73N (6b). The subpopulations that remain unchanged after the treatment with SKI-73 (6a) or SKI-73N (6b) (Subpopulation 1/7/12/15 of $G_0/G_1$-phase cells; 2/4/5 of S-phase cells; 1 of $G_2$/M-phase cells) were defined as the common resistant subset. SKI-73(6a)-specific emerging subpopulations (Subpopulation 3/4/5/6/16 of $G_0/G_1$-phase cells; 6 of S-phase cells; 4 of $G_2$/M-phase cells) are expected to be suppressed by CARM1 but emerge upon its inhibition. The remaining subpopulations are either associated with the effects of the small-molecule scaffold of SKI-73 (6a)/SKI-73N (6b) (commonly emerging Subpopulation 9 of $G_0/G_1$-phase cells, 0/5 of $G_2$/M-phase cells; commonly depleted Subpopulation 2/3 of $G_2$/M-phase cells) or with SKI-73N (6b)-specific effects (differentially depleted Subpopulation 10/18 of $G_0/G_1$-phase cells, 0 of S-phase cells; differentially emerging Subpopulation 20 of $G_0/G_1$-phase cells). Interestingly, scRNA-seq analysis of *CARM1*-KO cells (in comparison with SKI-73 (6a)-treated cells)suggests that *CARM1* knockout has more profound effects on the overall landscape of epigenetic plasticity (*Figure 7d*, *Figure 7—figure supplement 33*). Collectively, the chemical probe SKI-73 (6a) alters the epigenetic plasticity of MDA-MB-231 breast cancer cells via the combined effects of SKI-73(6a)'s molecular scaffold and specific inhibition of CARM1's methyltransferase activity.

## Identification of CARM1-dependent, invasion-prone subpopulations of breast cancer cells

Given that SKI-73 (6a) has no effect on the cell cycle and proliferation of MDA-MB-231 cells under the current treatment dose and duration, we envision that the invasion capability of MDA-MB-231 cells mainly arises from an invasion-prone subset, 80% of which is depleted by SKI-73 (6a) treatment (*Figure 6c–e*). We thus focused on Subpopulations 0/2/8/11/13/14/17/19 of $G_0/G_1$-phase cells and Subpopulation 3 of S-phase cells: in total nine depleted subpopulations specific for SKI-73 (6a) (*Figure 7c*, *Figure 7—figure supplements 31* and *32*, *Supplementary file 1*-Table F–H). To identify invasion-prone subpopulation(s) among these candidates, we compared the transcriptional signature (s) of these subpopulations with those of cells that freshly invaded through Matrigel within 16 hr. Strikingly, in comparison with the highly heterogenous scRNA-seq signature of the parental MDA-MB-231 cells, the freshly harvested invasive cells (3793 cells for scRNA-seq) are relatively homogeneous, with their subpopulations mainly determined by the cell-cycle-related transcriptomic signatures (*Figure 7d*, *Figure 7—figure supplements 33–37*). We then classified the freshly harvested invasive cells into $G_0/G_1$, S and $G_2$/M stages (*Figure 7—figure supplements 38–40*, *Supplementary file 1*-Table E). Through correlation analysis comparing the invasive cells and the subpopulations within each cell-cycle stage (*Figure 7d*, *Figure 7—figure supplements 39–42*), we revealed the subsets whose transcriptional signatures closely relate to those of the invasive cells, including Subpopulations 6/7/8/9/14 in $G_0/G_1$-phase cells, 0/3 in S-phase cells and 1/2 of $G_2$/M-phase cells (*Supplementary file 1*-Table F–K). In the context of population analysis for the nine SKI-73 (6a)-specific depleted subpopulations, Subpopulations 8/14 of $G_0/G_1$-phase cells and Subpopulation 3 in S-phase are putative invasion-prone candidates. Subpopulation 8 of $G_0/G_1$-phase cells is the most sensitive and the only subpopulation that can be depleted by around 80% with SKI-73 (6a) treatment (*Figure 7c*). Given the ~80% suppression and ~20% residual invasion capability upon SKI-73 (6a) treatment, we argue that the invasive phenotype of MDA-MB-231 cells predominantly arises from Subpopulation 8 $G_0/G_1$-phase cells, which only accounts for ~8% of the parental cells in $G_0/G_1$ phase (~5% without cell-cycle awareness). Differential expression analysis further revealed the single-cell transcriptional signatures of metastasis-implicated genes (*e.g.* MORC4,

S100A2, RPL39, IFI27, ARF6, CHD11, SDPR and KRT18) that are specific for the $G_0/G_1$-phase Subpopulation 8 and invasive cells but not for other $G_0/G_1$-phase invasion-prone candidates such as Subpopulation 6/7/9/14 (*Figure 7e*, *Figure 7—figure supplement 43*, *Supplementary file 1*-Table L). The remaining cells of $G_0/G_1$-phase Subpopulation 8 after **SKI-73** (**6a**) treatment (*Figure 7c,d*) together with others (subpopulation-6/7/9/14 in $G_0/G_1$-phase cells, 0/3 in S-phase cells and 1/2 of $G_2$/M-phase cells, *Figure 7—figure supplements 31*, *32*, *41*, *42*) may account for the 20% residual invasion capacity.

In the context of **SKI-73** (**6a**)-specific depletion of $G_0/G_1$-phase subpopulations, there are **SKI-73** (**6a**)-specific emerging $G_0/G_1$-phase subpopulations: Subpopulation 3/4/5/6/16 (*Figure 7c*). Population analysis of $G_0/G_1$-phase cells further revealed that Subpopulations 4 and 16 account for 90% of the emerging subset upon **SKI-73** (**6a**) treatment (*Supplementary file 1*-Table F). The transcriptional signatures and probably the associated invasion capability of Subpopulations 4 and 16 are dramatically different from those of the freshly harvested invasive cells and the bulk population of the parental cells, including the invasion-prone Subpopulation 8 (*Figure 7b,d*). Collectively, either *CARM1* knockout or CARM1 inhibition with **SKI-73** (**6a**) alters the epigenetic plasticity in a proliferation-independent manner by replacing the most invasion-prone subpopulation with the non-invasive subpopulation(s) to suppress the invasive phenotype (*Figure 7f*).

## Discussion

### Chemical probes of CARM1

On the basis of a novel small-molecule scaffold, 6'-homosinefungin (**HSF**), **SKI-73** (**6a**) was developed as a pro-drug-like chemical probe for CARM1 by cloaking the 9'-amine moiety of **5a** with the **TML** moiety. **SKI-73N** (**6b**) was developed as a control compound for **SKI-73** (**6a**). The inhibitory activity of **SKI-73** (**6a**) against CARM1 was demonstrated by the ability of **SKI-73** (**6a**) but not **SKI-73N** (**6b**) to abolish the cellular methylation marks of CARM1: the Arg1064 methylation of BAF155 and the Arg455/Arg460 methylation of PABP1 (*Lee and Bedford, 2002*; *Wang et al., 2014a*). The ready intracellular cleavage of **TML** is expected for the conversion of **SKI-73** and **SKI-73N** (**6a** and **6b**) into **5a** and **5b**, respectively, but it is remarkable that **SKI-73** and **SKI-73N** (**6a** and **6b**) can also be efficiently processed into **2a** and **2b** inside cells. In comparison, **6a** showed decent metabolic stability with no production of **2a** in the presence of microsomes. Here, **2a** and **5a** are presented as potent and selective CARM1 inhibitors, whereas their control compounds **2b** and **5b** interact poorly with CARM1.

Competitive assays with the SAM cofactor and the peptide substrate showed that **2a** and **5a** act on CARM1 in a SAM-competitive and substrate-noncompetitive manner. The SAM-competitive mode is consistent with the ligand–complex structures of CARM1, in which the SAM binding site is occupied by **2a** and **5a**. Strikingly, as revealed by their ligand–CARM1 complex structures, **2a** and **5a** engage CARM1 through noncanonical modes, with their 6'-*N*-benzyl moieties in the binding pocket that is otherwise occupied by the α-amino carboxylate moiety of conventional SAM analogs such as **SAH**, **SNF** and **1**. This observation is consistent with the 4.1–6.5 °C increase in the in vitro and cellular $T_m$ of CARM1 upon binding **2a** and **5a**, which contrasts with the smaller $T_m$ changes with SAM as a ligand. The distinct modes of interaction of CARM1 with **2a** and **5a** (*Figure 3c,g*) also rationalize the CARM1 selectivity of the two SAM analogs over other methyltransferases, including closely related PRMT homologs. Through mathematic modeling using the inputs of the LC-MS/MS-quantified intracellular concentrations and CARM1-binding constants of relevant **HSF** derivatives and the SAM cofactor, we concluded that high intracellular concentrations of **5a** and **2a**, and thus efficient CARM1 occupancy, can be achieved rapidly and maintained for several days with a single low dose of **SKI-73** (**6a**). By contrast, the occupancy by **5a** and **2a** of SMYD2, the next likely engaged target, is below 20% with the efficacious doses of **6a** that affect cell invasion. The polar α-amino acid zwitterion moiety of **2a** and the polar α-amino moiety of **5a** probably account for their accumulation and retention inside cells.

To the best of our knowledge, EZM2302, TP-064, and **SKI-73** (also **6a** in this work, www.thesgc. org/chemical-probes/SKI-73) and their derivatives are the only selective and cell-active CARM1 inhibitors (*Drew et al., 2017*; *Nakayama et al., 2018*). Although the potency, selectivity, on-target engagement and potential off-target effects associated with these compounds have been examined

in vitro and in cellular contexts as chemical probes, EZM2302, TP-064, and **SKI-73** (**6a**) are differentiated by their molecular scaffolds and modes of interaction with CARM1 (www.thesgc.org/chemical-probes/SKI-73) (*Drew et al., 2017*; *Nakayama et al., 2018*). **SKI-73** (**6a**) is a cofactor analog inhibitor embedding a *N6'*-homosinefungin moiety to engage the SAM binding site of CARM1 in a cofactor-competitive, substrate-noncompetitive manner; EZM2302 and TP-064 occupy the substrate-binding pocket of CARM1 in a SAH-uncompetitive or SAM-noncompetitive manner (*Drew et al., 2017*; *Nakayama et al., 2018*). In particular, the prodrug property of **SKI-73** (**6a**) allows its ready cellular uptake, followed by rapid conversion into its active forms inside cells. The prolonged intracellular CARM1 inhibition further distinguishes **SKI-73** (**6a**) from EZM2302 and TP-064.

## Anti-cancer effects and conventional mechanisms associated with pharmacological inhibition of CARM1

With **SKI-73** (**6a**) as a CARM1 chemical probe and **SKI-73N** (**6b**) as a control compound, we showed that pharmacological inhibition of CARM1 with **SKI-73** (**6a**), but not **SKI-73N** (**6b**), suppressed 80% of the invasion capability of MDA-MB-231 cells. By contrast, the pharmacological inhibition of CARM1 with **SKI-73** (**6a**) had no effect on the proliferation of MDA-MB-231 cells. This result is consistent with the lack of anti-proliferation activities of the other two CARM1 chemical probes, EZM2302 and TP-064, against breast cancer cell lines (*Drew et al., 2017*; *Nakayama et al., 2018*). The anti-invasion efficiency of **SKI-73** (**6a**) is in good agreement with the intracellular occupancy and the resulting abolition of several methylation marks of CARM1 upon treatment with **SKI-73** (**6a**). Our prior work showed that the methyltransferase activity of CARM1 is required for breast cancer metastasis (*Wang et al., 2014a*). Among the diverse cellular substrates of CARM1 (*Blanc and Richard, 2017*), BAF155—a key component of the SWI/SNF chromatin-remodeling complex–is essential for the invasion of MDA-MB-231 cells (*Wang et al., 2014a*). Mechanistically, the CARM1-mediated Arg1064 methylation of BAF155 facilitates the recruitment of the SWI/SNF chromatin-remodeling complex to a specific subset of gene loci (*Wang et al., 2014a*). Replacement of the native CARM1 with its catalytically dead mutant or with an Arg-to-Lys point mutation at the Arg1064 methylation site of BAF155 is sufficient to abolish the invasive capability of breast cancer cells (*Wang et al., 2014a*). CARM1 inhibition with **SKI-73** (**6a**), but not with its control compound **SKI-73N** (**6b**), recapitulates the anti-invasion phenotype associated with the genetic perturbation of CARM1. More importantly, there is no additive effect upon combining *CARM1*-KO with **SKI-73** (**6a**) treatment, underlying the fact that the two orthogonal approaches target the commonly shared pathway(s) that are essential for the invasion of breast cancer cells. In comparison to **SKI-73** (**6a**), the CARM1 inhibitors EZM2302 and TP-064 demonstrated anti-proliferation effects on hematopoietic cancer cells, in particular multiple myeloma (*Drew et al., 2017*; *Greenblatt et al., 2018*; *Nakayama et al., 2018*). Mechanistically, genetic perturbation of CARM1 in the context of leukemia impairs cell-cycle progression, promotes myeloid differentiation, and ultimately induces apoptosis, probably by targeting pathways of proliferation and cell-cycle progression, that is, E2F-, MYC-, and mTOR-regulated processes (*Greenblatt et al., 2018*). In comparison, CARM1 inhibition with EZM2302 led to a slightly different phenotype, which includes reduction of RNA stability, E2F target downregulation, and induction of a p53 response signature for senescence. (*Greenblatt et al., 2018*). Collectively, the effects of CARM1 chemical probes are highly context-dependent, with **SKI-73** (**6a**) having different uses in impairing the invasiveness of breast cancer cells, while TP-064 and EZM2302 have uses in preventing the proliferation of hematopoietic cancer cells.

## CARM1-dependent epigenetic plasticity revealed by SKI-73 (6a) with single-cell resolution

Given the increased awareness of epigenetic plasticity (*Flavahan et al., 2017*), we employed the scRNA-seq approach to examine MDA-MB-231 cells and their responses to chemical and genetic perturbation with CARM1. Because of the lack of a prior reference to define subpopulations of MDA-MB-231 cells, we developed a cell-cycle-aware algorithm to cluster the subpopulations with a resolution that was able to dissect subtle changes upon treatment with **SKI-73** (**6a**) versus its control compound **SKI-73N** (**6b**) in each cell-cycle stage. Guided by Silhouette analysis, the population entropy analysis and the Fisher Exact test, >10,000 MDA-MB-231 breast cancer cells were classified on the basis of their cell-cycle stages and then clustered into 34 subpopulations. With further

annotation of these subpopulations according to their different responses to treatment with **SKI-73** (**6a**) versus **SKI-73N** (**6b**), we readily dissected the subpopulations that were altered in a **SKI-73**(**6a**)-specific (CARM1-dependent) manner and then identified subsets with transcriptional signatures that are similar to that of the freshly isolated invasive cells. Quantitative analysis of **SKI-73** (**6a**)-depleted subpopulations further revealed the most invasion-prone subpopulation, which accounts for only 5% of the total population but at least 80% of the invasive capability of the parental cells. Collectively, we propose a model in which MDA-MB-231 cells consist of subpopulations, with their epigenetic plasticity (*Figure 7f*) determined by multiple factors including the CARM1-involved BAF155 methylation (*Wang et al., 2014a*). **SKI-73** (**6a**) inhibits the methyltransferase activity of CARM1, the Arg1064 methylation of BAF155, and thus the target genes associated with the methylated BAF155. These effects alter the cellular epigenetic landscape by affecting certain subpopulations of MDA-MB-231 cells without any apparent effect on cell cycle and proliferation. In the context of the invasion phenotype of MDA-MB-231 cells, the subset of invasion-prone cells is significantly suppressed upon the treatment with **SKI-73** (**6a**). Essential components that are used to dissect the invasion-prone population in this CARM1-dependent epigenetic plasticity model are the scRNA-seq analysis of sufficient MDA-MB-231 cells (>10,000 cells here), the utility of the freshly isolated invasive cells as the reference, the timing and duration of treatment, and the use of **SKI-73N** (**6b**) and DMSO as controls. Interestingly, although the invasion-prone subpopulation is also abolished in the *CARM1*-KO strain, *CARM1*-KO reshapes the epigenetic plasticity in a much more profound manner, significantly reducing the subpopulation heterogeneity of MDA-MB-231 cells. The distinct outcomes for the pharmacological and genetic perturbation could be due to their different modes of action: short-term treatment with **SKI-73** (**6a**) versus long-term clonal expansion of *CARM1*-KO cells. The pharmacological inhibition captures the immediate response, whereas the genetic perturbation reports long-term and potential resistant outcomes. This work thus presents a new paradigm to understand cancer metastasis in the context of epigenetic plasticity and provides guidance for similar analyses in broader contexts: other cell lines, patient-derived xenograft samples, and in vivo mouse models of breast cancer.

# Materials and methods

## Key resources table

| Reagent type (species) or resource | Designation | Source or reference | Identifiers | Additional information |
|---|---|---|---|---|
| Gene (*Homo sapiens*) | Human CARM1 catalytic domain | UniProtKB/Swiss-Prot: Q86 × 55.3 (positions 140–480) | | With an N-terminal 6 × His tag in pFBOH-MHL, for crystallization |
| Cell line (*Homo sapiens*, female) | MDA-MB-231 (female) | ATCC | | |
| Cell line (*Homo sapiens*, female) | MCF-7 (female) | ATCC | | |
| Cell line (*Homo sapiens*, female) | MDA-MB-231; CARM1-KO | *Wang et al., 2014a* | | |
| Cell line (*Homo sapiens*, female) | MCF-7; CARM1-KO | *Wang et al., 2014a* | | |
| Software, algorithm | REFMAC | PMID: 15299926 | | |
| Software, algorithm | SEquence Quality Control | https://github.com/ambrosejcarr/seqc.git | | |
| Software, algorithm | fisher.test | http://mathworld.wolfram.com/FishersExactTest.html | | |

*Continued on next page*

*Continued*

| Reagent type (species) or resource | Designation | Source or reference | Identifiers | Additional information |
|---|---|---|---|---|
| Software, algorithm | BuildClusterTree | https://rdrr.io/cran/Seurat | | |
| Software, algorithm | FindMarkers | https://rdrr.io/cran/Seurat | | |
| Software, algorithm | DoHeatmap | https://rdrr.io/cran/Seurat | | |

### Synthesis and characterization of 3, 1, 2a, 5a, 6a, 2b, 5b and 6b

#### Abbreviations in chemical structures
Ac, acetyl; Bn, benzyl; Bz, benzoyl; Cbz, benzyloxycarbonyl; TFA, trifluoroacetyl.

#### Abbreviations of chemical reagents
CbzCl, benzyl chloroformate; DCM, dichloromethane; DMF, $N$, $N$-dimethylformamide; HATU, 1-[bis (dimethylamino)methylene]$-$1 H-1,2,3-triazolo[4,5-b] pyridinium 3-oxid hexafluorophosphate; TFA, trifluoroacetic acid; THF, tetrahydrofuran; TML-NHS, $N$-hydroxysuccinimidyl ester 3-methyl-3-(2,4,5-trimethyl-3,6-dioxocyclo-hexa-1,4-dienyl)-butanoic acid; SAH, $S$-adenosyl homocysteine.

#### General experimental information
Reagents for chemical reactions were purchased from Sigma-Aldrich without purification unless mentioned otherwise. Anhydrous solvents were prepared from a solvent purification system (PURE SOLV, Innovative Technology, Inc). Chemical reactions were carried out in an argon atmosphere at the temperatures displayed by the thermocouple or at ambient temperature (22°C) unless described otherwise. The phrase "concentrated" in the synthetic method session refers to the reaction workup to remove volatile solvents that uses a rotary evaporator attached to a diaphragm pump (15–20 Torr) and then a high vacuum pump (<1 Torr). Chromatography purification was carried out with silica gel from Dynamic Adsorbents, Inc (neutral, 32–63 µm). NMR spectra were recorded on Bruker AVIII 600MHz spectrometers and reported in terms of chemical shifts (ppm), multiplicities (s = singlet, d = doublet, t = triplet, q = quartet, p=pentet, m = multiplet, and br = broad), and integration and coupling constants ($J$ in Hz). Chemical shifts were recorded with residual proton peaks of deuterated solvents as references (residual [1]H of DMSO, 2.50 ppm; CD$_3$OD, 3.31 ppm; D$_2$O, 4.80 ppm; [13]C of DMSO, 39.52 ppm; CD$_3$OD, 49.00 ppm). [1]H-NMR spectra were recorded at 24.0°C or at 70.0°C. [1]H-NMR spectra at 70.0°C were recorded in DMSO-$d_6$ to facilitate the equilibrium between rotamers. [13]C-NMR spectra were recorded at 24°C. Mass spectra for compound characterization were collected by Waters Acuity SQD LC-MS in electron spray ionization (ESI) mode. The final concentrations of the stock solutions of **3**, **1**, **2a**, **5a**, **5b**, SAH and nonradioactive SAM were determined on the basis of their UV absorption at 260 nm ($\varepsilon_{260}$ = 15,400 L.mol$^{-1}$.cm$^{-1}$) using a Nanodrop 1000 Spectrophotometer (Thermo Scientific). The final concentrations of **6a** and **6b** were determined with [1]H-NMR in CD$_3$OD containing 1.0 mM SAH as an internal reference for the first time and then with a Nanodrop 1000 Spectrophotometer (Thermo Scientific) on the basis of their UV absorption at 267 nm ($\varepsilon_{267}$ = 19,300 L.mol$^{-1}$.cm$^{-1}$) thereafter. CD$_3$-SAM was prepared as described previously (*Linscott et al., 2016*).

#### Synthesis of 3
Compound **3** was prepared as previously reported (*Wu et al., 2016*). Briefly, $N^6$-benzoyladenine (44 mg, 0.18 mmol), hexamethyldisilazane (3 mL) and pyridine (1 mL) were added into an oven-dried flask. The suspension was heated at 115°C to produce a clear solution and stirred for another 3 hr. The mixture was concentrated under reduced pressure. The residual was dried using azeotrope with toluene (3 × 5 mL) and over high vacuum. A solution of triacetate derivative (0.037 mmol) in 15 mL 1,2-dichloroethane to the solid crude of bis-sily-$N^6$-benzoyladenine. The resultant suspension was treated with TMSOTf (33 µL, 0.18 mmol) and heated at 50°C for 2 hr. The mixture was quenched with 10 mL saturated NaHCO$_3$, followed by extraction with 3 × 20 mL CH$_2$Cl$_2$. The combined

organic phases were washed with brine, dried with anhydrous $Na_2SO_4$, and evaporated to produce 56 mg crude of **3**. Flash silica gel chromatography with MeOH/DCM = 30:1 produced 30 mg of **3** as a white solid (92% yield).

**3**: $^1$H-NMR (600 MHz, DMSO-$d_6$, 84°C): δ 10.76(s, 1H), 9.39(d, 1H, $J$ = 6.4), 8.71(s, 1H), 8.59(s, 1H), 8.04(d, 2H, $J$ = 7.6), 7.63(t, 1H, $J$ = 7.4), 7.54(t, 2H, $J$ = 7.6), 7.22–7.31(m, 8H), 7.12–7.13(m, 2H), 6.23(d, 1H, $J$ = 5.6), 6.04(t, 1H, $J$ = 5.6), 5.41(t, 1H, $J$ = 5.6), 5.10(d, 2H, $J$ = 5.5), 4.45(d, 1H, $J$ = 15.6), 4.39(d, 1H, $J$ = 15.6), 4.24–4.28(m, 1H), 4.15–4.18(m, 1H), 3.66(s, 3H), 3.23–3.26(m, 1H), 3.13–3.16(m, 1H), 2.10(s, 3H), 2.04(s, 3H), 1.92–1.95(m, 1H), 1.81–1.88(m, 2H), 1.75–1.80(m, 1H), 1.64–1.67(m, 1H), 1.31–1.39(m, 2H); $^{13}$C-NMR (150 MHz, DMSO-$d_6$ *rotamers*): δ 170.92, 169.60, 169.51, 169.40, 165.67, 156.59(q, $J$ = 36.4), 155.87, 151.85, 151.80, 150.71, 143.88, 137.85, 136.77, 133.23, 132.57, 128.54, 128.52, 128.43, 128.32, 128.29, 127.79, 127.51, 127.39, 127.28, 127.12, 126.90, 126.01, 115.77(q, $J$ = 286.1), 85.85, 85.75, 79.17, 73.17, 71.89, 66.52, 66.38, 54.94, 52.82, 34.25, 32.81, 32.41, 30.72, 27.48, 26.70, 20.42, 20.38, 20.25; MS(ESI) m/z: 940 ([M+Na]$^+$; HRMS: calculated for $C_{45}H_{47}N_7O_{11}F_3$ ([M+H]$^+$)918.3286, found 918.3311.

## Synthesis of 1 and 2a from 3

To a solution of **3** (45 mg, 0.050 mmol) in 15 mL $CF_3CH_2OH$, we added Pd/C (22 mg, 10 wt. %, wet support, Degussa type). This mixture was stirred under $H_2$ (one atm) at ambient temperature (22°C) for 36 hr (*Bailey et al., 2008*). The reaction mixture was filtered through a short pad of Celite, followed by washing with 150 mL of MeOH. The combined filtrate was concentrated to produce around 1:1 crude mixture of **S1** and **S2** (*Figure 1—figure supplement 1*). This crude material was dissolved in 2 mL of MeOH and then mixed with 1 mL of 0.2 M LiOH. The resultant mixture was stirred at ambient temperature (22°C) for 40 hr, neutralized with 0.2 M HCl to reach pH = 7, and concentrated under reduced pressure to produce the crude product mixture of **1** and **2a**. The crude products were then subjected to a preparative reversed-phase HPLC (XBridge Prep C18 5 μm OBD 19 × 150 mm) with 5–95% gradient of $CH_3CN$ in aqueous trifluoroacetic acid (TFA, vol. 0.1%) in 24 min with a flow rate of 10 mL/min to afford 6.5 mg of **1** (27% yield over two steps) and 7.5 mg of **2a** (31% yield over two steps) as white solids. (See *Figure 1—figure supplement 1*).

**1**: $^1$H-NMR (600 MHz, $D_2O$): δ 8.36(s, 1H), 8.35(s, 1H), 6.01(d, 1H, $J$ = 3.7), 4.65 (dd, 1H, $J$ = 3.7, 5.4), 4.36(t, 1H, $J$ = 5.9), 4.22–4.19 (m, 1H), 4.03–3.99 (m, 1H), 3.82(t, 1H, $J$ = 6.0), 3.05 (dd, 1H, $J$ = 5.4 12.8), 2.96 (dd, 1H, $J$ = 6.9, 12.8), 1.97–1.84 (m, 5H), 1.49–1.55(m, 2H); $^{13}$C-NMR (150 MHz, $D_2O$): δ 173.33, 150.08, 148.17, 144.67, 142.76, 119.02, 115.28, 88.81, 81.18, 73.35, 53.79, 41.94, 33.65, 33.12, 26.88, 25.84; HRMS: calculated for $C_{16}H_{26}N_7O_5$ ([M+H]$^+$) 396.1995, found: 396.1982.

**2a**: $^1$H-NMR (600 MHz, $CD_3OD$): δ 8.16(s, 1H), 8.09(s, 1H), 7.25(d, 1H, $J$ = 7.4), 7.20(t, 2H, $J$ = 7.4), 7.12(t, 2H, $J$ = 7.4), 5.84(d, 1H, $J$ = 4.0), 4.49(dd, 1H, $J$ = 5.4, 4.0), 4.08 (t, 1H, $J$ = 5.7), 4.04 (d, 1H, $J$ = 13.0), 3.98–4.00(m, 1H), 3.95(d, 1H, $J$ = 13.0), 3.82(t, 1H, $J$ = 7.0), 2.94–3.00(m, 1H), 1.92–2.03(m, 2H), 1.83–1.91(m, 2H), 1.76–1.80(m, 1H), 1.46–1.56(m, 2H); $^{13}$C-NMR (150 MHz, $CD_3OD$): δ 171.94, 155.85, 151.31, 150.18, 142.68, 132.10, 130.90 (2×$^{13}$C), 130.83, 130.37 (2×$^{13}$C), 121.19, 91.65, 82.07, 75.36, 74.88, 54.07, 53.01, 52.07, 34.69, 34.32, 28.72, 27.90. HRMS: calculated for $C_{23}H_{32}N_7O_5$ ([M+H]$^+$) 486.2465, found:486.2464.

## Synthesis of 5a and 6a (SKI-73) from 3

To a solution of **3** (91 mg, 0.10 mmol) in 20 mL MeOH, we added 10 mL of 0.2 M LiOH. The resultant mixture was stirred at ambient temperature (22°C) for 40 hr. The reaction mixture was then neutralized with 0.2 M HCl to reach pH = 7.0 and concentrated under reduced pressure to produce **4** without further purification. The crude product **4** was then dissolved in 10 mL of THF and then mixed with 3 mL of saturated aqueous $NaHCO_3$ and CbzCl (12 μL, 0.10 mmol) at 0°C. This mixture was stirred at 0°C for 3 hr, quenched with 0.2 M HCl to reach pH = 7.0 and concentrated under reduced pressure. The resultant solid was washed with 100 mL THF, then filtered and concentrated to afford 65 mg of the crude product **S3** without further purification. To a solution of **S3** in 8 mL DMF, we sequentially added HATU (114 mg, 0.3 mmol), 2,3,5-collidine (39 μL, 0.3 mmol) and 4-methoxyphenethylamine (44 μL, 0.3 mmol) (*Han and Kim, 2004*). The resultant mixture was stirred at ambient temperature (22°C) under argon until the starting material **S3** was fully consumed as monitored by LC-MS. The reaction was then quenched with 3 mL of saturated aqueous $NH_4Cl$, followed by extraction with 3 × 30 mL DCM. The combined organic layers were washed with brine, dried with

anhydrous $Na_2SO_4$, filtered, and evaporated to give the crude solid product **S4**. This crude product was purified by a flash silica gel chromatography (v/v 1:12, MeOH/DCM) to give 55 mg of **S4** as a white solid (62% yield over three steps). (See *Figure 1—figure supplement 2*).

**S4**: [1]H-NMR (600 MHz, DMSO-$d_6$, 70°C): δ 8.28 (s, 1), 8.24 (s, 1), 7.53 (br, 1), 7.35–7.20 (m, 13), 7.11 (d, 2, J = 7.02), 7.07 (d, 2, J = 8.52), 6.81 (d, 2, J = 8.52), 5.86 (d, 1, J = 5.03), 5.10 (d, 1, J = 12.9), 5.08 (d, 1, J = 12.9), 5.03 (s, 2), 4.60 (t, 1, J = 5.03), 4.47 (d, 1, J = 16.2), 4.38 (d, 1, J = 16.2), 3.98 (t, 1, J = 5.03), 3.93–3.87 (m, 2), 3.69 (s, 3), 3.53–3.50 (m, 2), 3.16–3.10 (m, 2), 2.63 (t, 2, J = 7.14), 1.96–1.85 (m, 1), 1.71–1.58 (m, 3), 1.50–1.41 (m, 1), 1.39–1.32 (m, 1), 1.28–1.20 (m, 1). [13]C NMR (150 MHz, DMSO-$d_6$, rotamers): δ 172.05 (15), 157.93, 156.47, 156.41, 156.23, 153.04, 149.04, 141.88, 138.18, 137.26 (2 × C), 131.48, 129.53 (2 × C), 128.80 (2 × C), 128.72 (2 × C), 128.70 (2 × C), 128.67, 128.19, 128.13, 128.03, 127.98, 127.68 (2 × C), 127.55, 127.17, 119.29, 113.99 (2 × C), 88.48, 82.88 (d), 73.86, 73.17, 66.82 (d), 65.80, 55.38, 55.23, 50.40 (2 × C) 40.62 (2 × C), 35.16, 34.46, 33.72, 33.37, 29.25(d), 28.03 (d). HRMS: calculated for $C_{48}H_{55}N_8O_9$ ([M+H[+]]) 887.4092, found: 887.4082.

To a solution of **S4** (20 mg, 0.022 mmol) in 20 mL $CF_3CH_2OH$, we added Pd/C (10 mg, 10 wt. %, wet support, Degussa type). The resultant mixture was stirred under $H_2$ (1 atm) at ambient temperature (22°C) until the starting material **S4** was fully consumed as monitored by LC-MS. The reaction mixture was filtered through a short pad of Celite, followed by washing with 150 mL of MeOH. The combined filtrate was concentrated under reduced pressure to give the crude product **5a**. This crude material is further subject to a preparative reversed-phase HPLC (XBridge Prep C18 5 μm OBD 19 × 150 mm) with 5–95% gradient (volume ratio) of $CH_3CN$ in aqueous trifluoroacetic acid (TFA, vol. 0.1%) in 24 min with a flow rate of 10 mL/min to afford 5.4 mg of **5a** as a white solid (40% yield).

**5a**: [1]H-NMR (600 MHz, CD$_3$OD): δ 8.31 (s, 1), 8.21 (s, 1), 7.36 (t, 1, J = 7.32), 7.30 (t, 2, J = 7.32), 7.25 (d, 2, J = 7.32), 7.09 (d, 2, J = 8.58), 6.82 (d, 2, J = 8.58), 5.96 (d, 1, J = 3.84), 4.56 (dd, 1, J = 3.84, 5.28), 4.17–4.14 (m, 2), 4.09–4.03 (m, 2), 3.76–3.72 (m, 1), 3.78 (s, 3), 3.53–3.48 (m, 1), 3.40–3.34 (m, 1), 3.03–2.97 (m, 2), 2.78–2.69 (m, 2), 2.09–2.05 (m, 1), 2.05–1.99 (m, 1), 1.91–1.82 (m, 2), 1.74–1.68 (m, 1), 1.48–1.43 (m, 1), 1.41–1.35 (m, 1). [13]C NMR (150 MHz, CD$_3$OD): δ 169.72, 159.80, 154.63, 149.98, 149.65, 142.95, 132.08, 132.05, 130.81 (2 × C), 130.78 (2 × C), 130.69, 130.22 (2 × C), 121.00, 115.03 (2 × C), 91.51, 82.20, 75.30, 74.90, 57.63, 55.74, 54.32, 52.88, 42.17, 36.65, 34.59, 34.37, 29.72, 27.64. HRMS: calculated for $C_{32}H_{43}N_8O_5$ ([M+H[+]]) 619.3356, found: 619.3326.

To a solution of **5a** (6 mg, 0.01 mmol) in 1 mL anhydrous DMF was added Et$_3$N (12 μL, 0.05 mmol) and *N*-hydroxysuccinimidyl ester 3-methyl-3-(2,4,5-trimethyl-3,6-dioxocyclo-hexa-1,4-dienyl) butanoic acid (TML-NHS ester) (3.5 mg, 0.01 mmol) at 0°C (*Rohde et al., 2006*). The resultant mixture was stirred under argon for 8 hr, quenched with 2 mL of saturated aqueous NH$_4$Cl solution and then concentrated under reduced pressure. The resultant product **6a** was purified a preparative reversed-phase HPLC (XBridge Prep C18 5 μm OBD 19 × 150 mm) with 5–95% gradient (volume ratio) of $CH_3CN$ in aqueous trifluoroacetic acid (TFA, vol. 0.1%) in 24 min with a flow rate of 10 mL/ min to afford 2.8 mg of **6a** as a yellow solid (33% yield).

**6a**: [1]H-NMR (600 MHz, CD$_3$OD): δ 8.26 (s, 1), 8.19 (s, 1), 7.38–7.31 (m, 3), 7.26–7.24 (m, 2), 7.08 (d, 2, J = 8.40), 6.82 (d, 2, J = 8.40), 5.93 (d, 1, J = 4.50), 4.62 (t, 1, J = 4.50), 4.15–4.05 (m, 5), 3.76 (s, 3), 3.25–3.20 (m, 2), 3.05 (dd, 1, J = 6.00, 6.90), 2.99 (dd, 1, J = 6.00, 6.90), 2.90 (d, 1, J = 14.8), 2.79 (d, 1, J = 14.8), 2.67 (t, 2, J = 7.20), 2.08 (s, 3), 2.07–2.00 (m, 1), 2.08–1.98 (m, 2), 1.94 (s, 3), 1.91 (s, 3), 1.92–1.82 (m, 1), 1.78–1.72 (m, 1), 1.41 (s, 3), 1.39 (s, 3), 1.38–1.29 (m, 2). [13]C NMR (150 MHz, CD$_3$OD): δ 192.12, 188.69, 174.46, 173.71, 159.76, 155.11, 154.3, 149.97, 149.24, 144.93, 143.15, 138.81, 138.71, 132.23, 132.09, 130.87 (2 × C), 130.83 (2 × C), 130.76, 130.27 (2 × C), 120.99, 114.97 (2 × C), 91.30, 82.60, 75.32, 74.89, 55.70, 54.22, 52.86, 51.80, 49.58, 42.09, 39.56, 35.52, 35.35, 33.99, 29.58, 29.27, 29.21, 28.70, 14.42, 12.93, 12.04. HRMS: calculated for $C_{46}H_{59}N_8O_8$ ([M+H[+]]) 851.4456, found: 851.4431.

## Synthesis of 5b and 6b (SKI-73N) from S5

Compound **S5** was prepared as described previously (*Zheng et al., 2012*). Into a solution of **S5** (94 mg, 0.10 mmol) in 20 mL MeOH solution, we added 10 mL of 0.2 M LiOH. The resultant mixture was stirred at ambient temperature (22°C) for 20 hr. The mixture was then neutralized with 0.2 M HCl to reach pH = 7 and concentrated under reduced pressure to afford the crude product **S6** without

further purification. To a solution of **S6** in 8 mL anhydrous DMF, we sequentially added HATU (114 mg, 0.3 mmol), 2,3,5-collidine (39 μL, 0.3 mmol) and 4-methoxyphenethylamine (44 μL, 0.3 mmol). The resultant mixture was stirred at ambient temperature (22°C) under argon until the starting material **S6** was fully consumed as monitored by LC-MS. The reaction was then quenched with 3 mL of saturated aqueous $NH_4Cl$, followed by the extraction with $3 \times 30$ mL DCM. The combined organic layers were washed with brine, dried with anhydrous $Na_2SO_4$, filtered, and evaporated to give the crude product **S7**. This crude product was purified by a flash silica gel chromatography (v/v 1:12, MeOH/DCM) to afford 61 mg of **S7** as a white solid (71% yield). (See *Figure 1—figure supplement 3*).

**S7**: [1]H-NMR (600 MHz, DMSO-$d_6$, 70°C): δ 8.29 (s, 1), 8.24 (s, 1), 7.50–7.47 (br, 1), 7.37–7.33 (m, 2), 7.28–7.20 (m, 12), 7.19–7.17 (m, 1), 7.07 (d, 2, J = 8.58), 6.81 (d, 2, J = 8.58), 5.84 (d, 1, J = 5.04), 5.07 (s, 2), 5.03 (d, 1, J = 12.7), 5.00 (d, 1, J = 12.7), 4.61 (t, 1, J = 5.04), 4.42 (d, 1, J = 15.8), 4.32 (d, 1, J = 15.8), 3.94 (t, 2, J = 5.04), 3.84–3.81 (m, 2), 3.70 (s), 3.25–3.20 (m, 2), 2.60 (t, 2, J = 7.14), 2.09–2.01 (m, 1), 1.85–1.82 (m, 1), 1.67–1.58 (m, 1), 1.56–1.46 (m, 2), 1.42–1.33 (m, 1). [13]C NMR (150 MHz, DMSO-$d_6$, *rotamers*): δ 171.45 (d), 157.57, 156.55, 155.81, 155.77, 152.22 (d), 148.55, 148.60, 141.95, 139.11, 137.01, 136.69 (d), 131.16, 129.53 (2 × C), 128.30 (2 × C), 128.22 (2 × C), 128.18 (2 × C), 128.08 (2 × C), 127.76, 127.64 (2 × C), 127.26, 127.05, 126.96, 126.71, 119.10, 113.62 (2 × C), 88.06, 81.56 (d), 73.52 (2 × C), 73.09 (2 × C), 66.47 (d), 65.35, 54.87, 54.84, 54.65, 40.38, 36.66 (d), 34.17, 29.47, 29.33. HRMS: calculated for $C_{47}H_{53}N_8O_9$ ([M+H+]) 873.3936, found: 873.3928.

To a solution of **S7** (20 mg, 0.022 mmol) in 20 mL $CF_3CH_2OH$, we added Pd/C (10 mg, 10 wt. %, wet support, Degussa type). This mixture was stirred under $H_2$ (1 atm) at ambient temperature (22°C) until the starting material **S7** was fully consumed as monitored by LC-MS. The reaction mixture was then filtered through a short pad of Celite, followed by washing with 150 mL of MeOH. The combined filtrate was concentrated under reduced pressure and purified with a preparative reversed-phase HPLC (XBridge Prep C18 5 μm OBD 19 × 150 mm) with 5–95% gradient (volume ratio) of $CH_3CN$ in aqueous trifluoroacetic acid (TFA, vol. 0.1%) in 24 min with a flow rate of 10 mL/min to afford 5.4 mg of **5b** as a white solid (40% yield).

**5b**: [1]H-NMR (600 MHz, CD₃OD): δ 8.28 (s), 8.14 (s), 7.30 (t, 1, J = 7.50), 7.21 (t, 2, J = 7.50), 7.13 (d, 2, J = 8.58), 7.08 (d, 2, J = 7.50), 6.83 (d, 2, J = 8.58), 5.96 (d, 1, J = 3.96, H1), 4.69 (dd, 1, J = 3.96, 5.88), 4.40 (t, 1, J = 5.88), 4.18–4.21 (m, 1), 4.10 (d, 1, J = 12.84), 4.05 (d, 1, J = 12.84), 3.85–3.83(m, 1), 3.72 (s, 3), 3.47–3.43 (m, 3), 2.79–2.74 (m, 2), 2.36 (ddd, 1, J = 4.32, 9.42, 15.1), 2.25 (ddd, 1, J = 3.00, 6.00, 15.1), 1.95–1.89 (m, 2), 1.85–1.80 (m, 2). [13]C NMR (150 MHz, CD₃OD): δ 169.29, 159.87, 156.45, 152.48, 150.06, 142.33, 132.00, 130.74 (2 × C), 130.58, 130.44 (2 × C), 130.10 (3 × C), 121.06, 115.05 (2 × C), 91.72, 80.78, 74.63, 74.12, 56.94, 55.68, 53.93, 49.50, 42.37, 35.47, 32.25, 28.71, 26.46. HRMS: calculated for $C_{31}H_{41}N_8O_5$ ([M+H+]) 605.3200, found: 605.3180.

To a solution of **5b** (6 mg, 0.01 mmol) in 1 mL anhydrous DMF, we added Et₃N (12 μL, 0.05 mmol) and *N*-hydroxysuccinimidyl ester 3-methyl-3-(2,4,5-trimethyl-3,6-dioxocyclo-hexa-1,4-dienyl) butanoic acid (TML-NHS ester) (3.5 mg, 0.01 mmol) at 0°C. The resultant mixture was stirred under argon for 8 hr, quenched with 2 mL of saturated aqueous $NH_4Cl$ solution and then concentrated under reduced pressure. The resultant crude product was purified a preparative reversed-phase HPLC (XBridge Prep C18 5 μm OBD 19 × 150 mm) with 5–95% gradient (volume ratio) of $CH_3CN$ in aqueous trifluoroacetic acid (TFA, vol. 0.1%) in 24 min with a flow rate of 10 mL/min to afford 3.0 mg of **6b** (**SKI-73N**) as a yellow solid (35% yield).

## 6b (SKI-73N)

[1]H-NMR (600 MHz, CD₃OD): δ 8.31 (s, 1), 8.18 (s, 1), 7.32 (t, 1, J = 7.38), 7.24 (t, 2, J = 7.38), 7.11–7.07 (m, 4), 6.82 (d, 2, J = 8.40), 5.96 (d, 1, J = 3.78), 4.69 (dd, 1, J = 3.78, 5.94), 4.35 (t, 1, J = 5.94), 4.19–4.16 (m, 2), 4.13 (d, 1, J = 12.9), 4.04 (d, 1, J = 12.9), 3.74 (s, 3), 3.44–3.40 (m, 1), 3.37–3.30 (m, 2), 2.92 (d, 1, J = 14.9), 2.78 (d, 1, J = 14.9), 2.68 (t, 2, J = 7.08), 2.38–2.32 (m, 1), 2.25–2.20 (m, 1), 2.08 (s, 3), 1.94 (s, 3), 1.91 (s, 3), 1.82–1.78 (m, 2), 1.74–1.70 (m, 1), 1.57–1.54 (m, 1), 1.40 (s, 6). [13]C NMR (150 MHz, CD₃OD): δ 192.14, 188.66, 174.51, 173.17, 159.79, 155.66, 155.10, 151.29, 150.04, 144.95, 142.82, 138.79, 138.73, 132.22, 130.79 (3 × C), 130.61, 130.43 (2 × C), 130.13 (2 × C), 121.07, 114.98 (2 × C), 91.65, 81.14, 74.84, 74.24, 57.62, 55.68, 53.77, 49.50 (2 × C), 42.18, 39.57, 35.56, 32.85, 29.29, 29.17, 29.10, 27.76, 14.42, 12.94, 12.04. HRMS: calculated for $C_{45}H_{57}N_8O_8$ ([M+H+]) 837.4299, found: 837.4316.

## Determination of the $IC_{50}$ of SAH, SNF, 1, 2a, 5a, 5b and 6a

The $IC_{50}$ assays of **SAH**, **SNF**, **1**, **2a**, **5a** and **5b** against a collection of 34 methyltransferases were performed as previously described except for PRMT9 (*Bromberg et al., 2017*; *Li et al., 2016a*; *Li et al., 2016b*). The $IC_{50}$ values were obtained by fitting %inhibition against the concentrations of the inhibitor to a sigmoid curve.

Briefly, the effects of these compounds on G9a, GLP1, SUV39H1, SUV39H2, SUV420H1, SUV420H2, SETD2, SETD8, SETDB1, SETD7/9, a trimeric complex of MLL1, a pentameric complex of MLL3, a trimeric complex of EZH2 (PRC2), PRMT1, PRMT3, CARM1, a PRMT5/MEP50 complex, PRMT6, PRMT7, PRMT8, PRMT9, PRDM9, SMYD2, SMYD3, DNMT1, BCDIN3D and METTL3-METTL14 were assessed by monitoring the incorporation of the tritium-labeled methyl group of [$^3$H-Me]-SAM into substrates using a scintillation proximity assay (SPA). Here, we prepared a 10 μl reaction containing [$^3$H-Me]-SAM and a substrate at their concentrations close to the apparent $K_m$ values for each enzyme. The reaction was quenched by adding 10 μl of 7.5 M guanidine hydrochloride. Into this mixture, we added 180 μL of 20 mM Tris buffer (pH 8.0). This mixture was then transferred to a 96-well FlashPlate followed by incubation for 1 hr. The counts per minute (CPM) was measured with a TopCount plate reader. The CPM readouts in the absence of these compounds or these enzymes were defined as 100% activity and background (0%), respectively. For BCDIN3D and METTL3-METTL14, biotinylated RNA strands were used as substrates. For DNMT1, the substrate was a double-stranded DNA with the forward strand biotinylated. For PRMT9, the assay buffer contains 10 nM PRMT9 (aa 1–845), 20 μM SAM, 80 nM biotinylated-SAP145 peptide substrate, 20 mM Tris-HCl (pH = 7.5), 5 mM DTT, and 0.01% Triton X-100. Purification of PRMT9 will be reported elsewhere.

A filter-based assay was applied for DOT1L, NSD1, NSD2, NSD3, ASH1L, DNMT3A/3L, and DNMT3B/3L. Here, we incubated a reaction mixture of 10 μL at 23°C for 1 hr, followed by adding 50 μL of 10% trichloroacetic acid (TCA) to quench the reaction. The resulting mixture was then transferred to filter plates (Millipore, Billerica, MA, USA), followed by centrifugation at 2000 rpm (Allegra X-15R; Beckman Coulter, Brea, CA, USA) for 2 min, washing twice with 10% TCA and once with 180 μL ethanol. After centrifugation and drying, 100 μL MicroScint-O (Perkin Elmer) was added into each well and the plates were centrifuged to remove the liquid. A 70 μL volume of MicroScint-O was added and the CPM was measured with a TopCount plate reader.

The radiometric filter paper assay described previously (*Ibanez et al., 2012*; *Zheng et al., 2012*) was also performed as the alternative assay to determine the $IC_{50}$ values of **2a**, **5a** and **6a** against CARM1. Briefly, 20 μL enzymatic reactions (triplicate of each data point) were carried out in the assay solution containing 50 mM HEPES-HCl (pH = 8.0), 0.005% Tween-20 (v/v), 0.0005% BSA, 1 mM TCEP, 25 nM CARM1, 1.5 μM H3 peptide (aa 1–40), 0.75 μM [$^3$H-Me]-SAM (PerkinElmer, 5–15 Ci/mmol in 9:1 v/v of sulfuric acid and ethanol), and the varied concentrations of inhibitors. For the signal readout of no inhibition (high readout), the same reaction (in triplicate) was carried out except with inhibitors replaced with DMSO. For the signal readout of 100% inhibition (background readout), the same reaction (in triplicate) was carried out except with inhibitors replaced with DMSO and with the omission of 25 nM CARM1. Here, we incubated a 10 μL solution containing 50 nM CARM1 and an examined inhibitor at 2 × concentrations at 22°C for 30 min. To the 10 μL mixture of CARM1 and an inhibitor, we added 10 μL of the assay buffer containing 3.0 μM H3 peptide (aa 1–40) and 1.5 μM [$^3$H-Me]-SAM. The resultant reaction was carried out at ambient temperature (22°C) for 6 hr, which is in the linear range of the increase of signal readouts. The reactions were then quenched by spotting the reaction mixture onto P81 ion-exchange cellulose chromatography paper (Fisher) to immobilize the peptide. The filter paper was washed 3 times with 50 mM $NaHCO_3$/$Na_2CO_3$ buffer (pH = 9.2) to remove the unreacted [$^3$H-Me]-SAM. The immobilized peptide containing [$^3$H-Me] was quantified by a scintillation counter (Tri-Carb 2810TR, PerkinElmer). To determine the $IC_{50}$ value of an inhibitor against CARM1, %inhibition at specific concentrations was calculated as [(CPM of high readout – CPM readout in the presence of the inhibitor)/(CPM of high readout – CPM of background readout)]×100%. The $IC_{50}$ values were obtained by fitting %inhibition against the concentrations of the inhibitor to a sigmoid curve with GraphPad Prism.

## Determination of $IC_{50}$ and $K_d$ values of CARM1 inhibitors and $K_{m,SAM}$ in the presence of the varied concentrations of SAM and peptide substrate

SAM- and substrate-dependent $IC_{50}$ values were determined with the radiometric filter paper assay as described above except in the presence of the varied concentrations of the SAM cofactor and the peptide substrate. The $IC_{50}$ values of **2a** and **5a** were obtained in the presence of 0.09, 0.18, 0.37, 0.75, 1.50, 2.25, 3.00, 5.62, and 7.50 µM [$^3$H-Me]-SAM or 0.3, 1.5, 7.5, 15, 30, and 50 µM H3 peptide (aa 1–40). The resultant $IC_{50}$ values were plotted as a function of the concentrations of SAM and the peptide substrate using GraphPad Prism Software. Given the SAM-competitive character of **2a** and **5a**, their SAM-dependent $IC_{50}$ values were then fitted according to *Equation 1*. Here [SAM] is the concentration of SAM, $K_{m,SAM}$ is the Michaelis-Menten constant, and $K_d$ is the dissociation constant of the CARM1 inhibitor **2a** or **5a**. $K_d$, $K_d/K_{m,SAM}$ and $K_{m,SAM}$ can therefore be obtained through the intercept of the y-axis, the slope, and their ratio, respectively, upon fitting *Equation 1*. Given that **6a** showed a much higher $IC_{50}$ (1.1 ± 0.1 µM, *Figure 5—figure supplement 3*), the $K_d$ value of **6a** ($K_{d,6a}$ = 0.32 µM) was directly obtained from the $IC_{50}$ value, the $K_{m,SAM}$ derived above, and the assayed SAM concentration according to *Equation 1*.

$$IC_{50} = [SAM] \times k_d/k_{m,SAM} + k_d \tag{1}$$

## Surface plasmon resonance (SPR)

Full-length CARM1 was used for SPR assay. DNA fragment encoding the full-length CARM1 was cloned into pFB-N-flag-LIC donor plasmid. The resulting plasmid was transformed into DH10Bac-competent *E. coli* cells (Invitrogen) and a recombinant Bacmid DNA was purified, followed by a recombinant baculovirus generation in Sf9 insect cells. Sf9 cells grown in HyQ SFX insect serum-free medium (ThermoScientific) were infected with 10 mL of P3 viral stocks per 1 L of suspension cell culture and incubated at 27°C using a platform shaker set at 150 revolutions per minute. The cells were collected when viability dropped to 70–80% (~72 hr after infection). Harvested cells were re-suspended in PBS, 1X protease inhibitor cocktail (100X protease inhibitor stock in 70% ethanol containing 0.25 mg/mL aprotinin, 0.25 mg/ml leupeptin, 0.25 mg/mL pepstatin A and 0.25 mg/mL E-64) and 2X Roche complete EDTA-free protease inhibitor cocktail tablet. The cells were lysed chemically by rotating 30 min with NP40 (final concentration of 0.6%) and 50 U/mL benzonase nuclease (Sigma), 2 mM 2-mercaptoethanol and 10% glycerol followed by sonication at frequency of 7 (10''on/10''off) for 2 min (Sonicator 3000, Misoni). The crude extract was clarified by high-speed centrifugation (60 min at 36,000 × g at 4°C) by Beckman Coulter centrifuge. The recombinant protein was purified by incubating the cleared lysate with anti-FLAG M2 affinity agarose gel (Sigma, Cat # A2220) and then rotating for 3 hr, followed by washing with 10 CV TBS (50 mM Tris-HCl, 150 mM NaCl, pH 7.4) containing 2 mM 2-mercaptoethanol, 1X protease inhibitor cocktail (100X protease inhibitor stock in 70% ethanol containing 0.25 mg/mL aprotinin, 0.25 mg/mL leupeptin, 0.25 mg/mL pepstatin A and 0.25 mg/mL E-64) and 1X Roche complete EDTA-free protease inhibitor cocktail tablet. The recombinant protein was eluted by competitive elution with a solution containing 100 µg/mL FLAG peptide (Sigma, Catalog # F4799) in 20 mM Tris pH:7.4, 150 mM NaCl, 5% glycerol, 3 mM 2-mercaptoethanol. Quality of CARM1 (>95%) was determined by SDS–PAGE. The protein was then concentrated, flash frozen with liquid nitrogen, and stored at −80°C for future use.

SPR analysis was performed using a Biacore T200 (GE Health Sciences Inc) at 25°C. Approximately 5500 response units of CARM1 (amino acids 1–608) were amino-coupled onto a CM5 chip in one flow cell according to the protocol of the manufacturer. Another flow cell was left empty for reference subtraction. SPR analysis was conducted in the HBS-EP buffer (20 mM HEPES pH 7.4, 150 mM NaCl, 3 mM EDTA, 0.05% Tween-20) containing 2% (v/v) DMSO. The stock solutions of five concentrations of compound **2a** (24.7, 74.1, 222, 667 and 2000 nM) and four concentrations of **5a** (6.2, 18.5, 55.6 and 167 nM) were prepared by serial dilution. Binding kinetic experiments were performed with single cycle kinetics with the contact time of 60 s and off time of 300 s in a flow rate of 30 µL/min. To facilitate complete dissociation of compound for the next cycle, a regeneration step (300 s, 40 µL/min of buffer), a period of stabilization (120 s) and two blank cycles were included between each cycle. Kinetic curve fittings were carried out with a 1:1 binding model or a heterogeneous ligand model using Biacore T200 Evaluation software (GE Health Sciences Inc).

## In vitro thermal shift assay (TSA)

TSA was performed as described previously (*Blum et al., 2014*; *Niesen et al., 2007*) to examine the melting temperature ($T_m$) of CARM1 in the presence or absence of ligands. For each measurement (triplicate of each data point), the assay solution containing 50 mM HEPES-HCl (pH = 8.0), Tween-20 0.005% (v/v), 1 mM TCEP, 0.5 μM CARM1, and 5 μM ligand (SAM, **1**, **2a** or **5a**) was mixed with 5 × SYPRO Orange Protein Gel Stain stock (Sigma Aldrich) in a 96-well PCR plate. The mixture was equilibrated at 25°C in dark for 5 min and then loaded onto Bio-Rad CFX96 Real-Time PCR Detection System. The fluorescence readouts were recorded upon increasing the heating temperature from 25°C to 100°C at a rate of 0.2 °C/s. The raw data of the fluorescence readouts *versus* the temperatures were exported with CFX software and processed as the percentage of the fluorescent signal normalized between the lowest readout of 0% and the highest readout of 100% within the 25–100°C region. The melting curves were plotted as the normalized fluorescence (%) versus the heating temperature and fit with a sigmoid curve with GraphPad Prism. The $T_m$ corresponds to the temperature with the 50% relative fluorescent signal in the sigmoidal curve.

## Crystallization of CARM1 in complex with 1 and 5a

A DNA fragment encoding the methyltransferase domain of human CARM1 (residues 140–480) was cloned into a baculovirus expression vector pFBOH-MHL (http://www.thesgc.org/sites/default/files/toronto_vectors/pFBOH-MHL.pdf). The protein was expressed in Sf9 cells as an N-terminal 6 × His tag fusion protein and purified by metal chelating affinity chromatography (TALON resin, Clontech, Mountain View, CA, USA) followed by size-exclusion chromatography (Superdex 200, GE Healthcare). Pooled fractions containing CARM1 were subjected to the treatment of tobacco etch virus to remove the 6 × His tag. The protein was purified to homogeneity by ion-exchange chromatography. Purified CARM1 (6.5 mg/mL) was crystallized with the sitting drop vapor diffusion method at 20°C.

For the CARM1–**1** complex, CARM1 was mixed with **1** at a 1:5 molar ratio (protein:ligand) and crystallized with the sitting drop vapor diffusion method at 20°C by mixing 1 μL of the protein solution with 1 μL of the reservoir solution containing 20% PEG3350 and 0.2 M diammonium tartrate. X-ray diffraction data for the CARM1–**1** complex were collected at 100 K at beam line 23ID-B of Advanced Photon Source (APS), Argonne National Laboratory. Data sets were processed using the HKL-2000 suite (*Otwinowski and Minor, 1997*). The structure of the CARM1–**1** complex was solved by molecular replacement using MOLREP (*Otwinowski and Minor, 1997*) with the PDB entry 2V74 as the search template. REFMAC (*Emsley and Cowtan, 2004*; *Murshudov et al., 1997*) was used for the structure refinement. Graphics program COOT was used for model building and visualization. Crystal diffraction data and refinement statistics for the structure are displayed in *Table 2*. To further confirm the electronic densities of the ligand, the total omission electron density map was calculated using SFCHECK from CCP4suite and contoured at 1.0 sigma (*Figure 4—figure supplement 1*) (*Emsley and Cowtan, 2004*; *Murshudov et al., 1997*).

For the CARM1-**5a** complex, CARM1 was crystallized with the sitting drop vapor diffusion method at 20°C by mixing 1 μL of protein solution with 1 μL of the reservoir solution containing 25% PEGG3350, 0.1M ammonium sulfate and 0.1 M Hepes (pH = 7.5). The compound **5a** (0.2 μL of 10 μM in DMSO) was added to the drops with apo crystals and incubated overnight. X-ray diffraction data for the CARM1-**5a** complex were collected at 100 K at beamline 24ID-E of Advanced Photon Source (APS), Argonne National Laboratory. Data were processed using the HKL-3000 suite (*Minor et al., 2006*). The structure was isomorphous to PDB entry 4IKP, which was used as a starting model. REFMAC (*Murshudov et al., 1997*) was used for structure refinement. Geometric restraints for compound refinement were prepared with GRADE v.1.102 developed at Global Phasing Ltd. (Cambridge, UK). The COOT graphics program (*Emsley et al., 2010*) was used for model building and visualization, and MOLPROBITY (*Williams et al., 2018*) was used for structure validation. To further confirm the electronic densities of the ligand, the total omission electron density map was calculated and contoured at 2.5 sigma (*Figure 3b*) (*Emsley and Cowtan, 2004*; *Murshudov et al., 1997*).

## Molecular docking and molecular dynamics simulations of CARM1–2a and CARM1–SNF complexes

The ligands were docked into the binding site of CARM1 using the induced-fit docking (IFD) protocol (*Sherman et al., 2006*) implemented in the Schrodinger suite (release 2016–4). The poses for **SNF** and **2a** were selected according to the IFD scores. Specifically, the results identified two distinct poses for **2a** with similar scores but only one pose for **SNF**. The poses were then further relaxed by all-atom, explicit solvent molecular dynamics (MD) simulations. Herein, CARM1 models in complex with the two ligands were placed into explicit water boxes. A simple point charge (SPC) water model (*Berendsen et al., 1981*) was used to solvate the system, charges were neutralized, and 0.15 M NaCl was added. The total system size was ~50,000 atoms. Desmond MD systems (D. E. Shaw Research, New York, NY) with OPLS3 force field (*Harder et al., 2016*) were used. The system was initially minimized and equilibrated with restraints on the ligand heavy atoms and protein backbone atoms, followed by production runs with all atoms unrestrained. The isothermal-isobaric ensemble was used with constant temperature (310 K) maintained with Langevin dynamics and 1 atm constant pressure achieved using the hybrid Nose-Hoover Langevin piston method (*Feller et al., 1995*) on a flexible periodic cell. For each CARM1–ligand complex, a 600 ns trajectory was collected.

## Cell lines

MCF-7 and MDA-MB-231 (parental and CARM1-*KO*) cell lines (*Wang et al., 2014a*) were used after their quality was confirmed with STR profile and standard mycoplasma contamination testing.

## LC-MS/MS quantification of intracellular concentrations of 6a, 5a, 2a and SAM

### Sample preparation for LC-MS/MS analysis

To measure the intracellular concentrations of **6a**, **5a**, **2a**, and SAM, a LC-MS/MS quantification method was developed with modification of what was reported previously (*Wang et al., 2014b*). Briefly, $0.2 \times 10^6$ MDA-MB-231 cells were incubated with varied concentrations of **6a** (0.5, 2.5, 5.0, and 10.0 µM) for several different periods of time (0.1, 3, 6, 12, 24, and 48 hr). The treated cells (in triplicate for each data point) were harvested and centrifuged at 4°C at 94 x g for 5 min. The cell pellets were resuspended and then washed with $4 \times 1$ mL of 4°C PBS to remove extracellular **6a**. The $4 \times 1$ mL washing is sufficient to remove extracellular **6a**, as evidenced by the LC-MS/MS analysis in which the residual **6a** in each washing buffer gradually decreased until the full loss of its MS signal after the 4 times of washing. The washed cell pellets were then treated with 40 µL MeOH (with 0.1% TFA, v/v) containing 0.125 µM **6b**, 0.125 µM **5b**, 16 µM **2b**, and 2 µM CD$_3$-SAM (*Linscott et al., 2016*) as the LC-MS/MS internal standards. The mixture was lysed by 150 W sonication at 0°C for 20 min. The resultant cell lysis was centrifuged with 21,130 g at 4°C for 20 min. The 30 µL aliquot of the MeOH extraction of each sample was collected into a 96-well plate (5042–1386, Agilent) and stored at −20°C until used for LC-MS/MS analysis.

### LC-MS/MS conditions

Liquid chromatography-tandem mass spectrometry analysis was performed with a 6410 triple-quad LC-MS/MS system (Agilent Technologies) in electrospray ionization (ESI) mode, equipped with an Agilent Zorbax Eclipse XDB-C18 column (2.1 × 50 mm, 3.5 µM). The samples were eluted with a 5–95% gradient (v/v) of CH$_3$CN in aqueous formic acid (HCOOH, 0.1%, v/v) in 7 min with a flow rate of 0.4 mL/min. The 96-well sample plate obtained above was maintained in the 4°C chamber of the LC-MS/MS prior to analysis. The 7 µL MeOH extraction of each sample was injected into the LC-MS/MS, and the MS signals were collected using the multiple-reaction monitoring (MRM) mode.

### Working curves of 6a, 5a, 2a, and SAM with 6b, 5b, 2b, and CD$_3$-SAM as internal standards

(See Figure 5c,e, Figure 5—figure supplements 1 and 2.) Standard working curves to quantify **6a**, **5a**, **2a**, and SAM were generated by plotting a linear function (*Equation 2*) with the ratio of the mass peak areas ($P_A/P_{IS}$) on the x-axis and the ratio of the concentration ($C_A/C_{IS}$) between each analyte (*A*) and the structurally related internal standard (*IS*) on the y-axis. To obtain the values of '*p*' and '*q*' in *Equation 2*, the 0.1% TFA MeOH (v/v) samples containing the varied concentrations of an

analyte (**6a**, **5a**, **2a** or SAM) and the fixed concentrations of the mixture of the internal standards (0.125 µM **6b**, 0.125 µM **5b**, 16 µM **2b**, and 2 µM CD$_3$-SAM) were subject to LC-MS/MS analysis. For **6a**, **5a**, and **2a**, three working curves for each analyte (a total of 9 for the three analytes) were generated to cover the concentration range of 3.9 nM–18.0 µM of the analyte ($C_{A,6a}$, $C_{A,5a}$ and $C_{A,2a}$) with the structurally related **6b**, **5b** and **2b** ($C_{IS,6b}$, $C_{IS,5b}$ and $C_{IS,2b}$) as internal standards. For SAM, one working curve was generated to cover the concentration range of 0.28–18 µM of SAM ($C_{A,SAM}$) with CD$_3$-SAM as the internal standard ($C_{IS,CD3-SAM}$).

$$Y = p \times X + q \tag{2}$$

## Quantification of intracellular concentrations of 6a, 5a, 2a, and SAM

Using the standard working curves generated above, the concentration ($C_A$) of each analyte (**6a**, **5a**, **2a**, and **SAM**) in the MeOH extraction of cell lysates was obtained through the ratio ($P_A/P_{IS}$) of the mass peak areas of each analyte ($P_A$) versus each internal standard ($P_{IS}$), and the concentration of the internal standard ($C_{IS}$) according to *Equation 2*. For **6a**, **5a**, and **2a** under each assay condition, three similar $C_A$ values ($C_{A-6b}$, $C_{A-5b}$, and $C_{A-2b}$) were obtained with **6b**, **5b** and **2b** as the internal standards, respectively. An average concentration of the analyte ($\overline{C}_A$) was obtained on the basis of the three concentrations weighted by the mass peak areas of the three internal standards ($P_{IS,6b}$, $P_{IS,5b}$ and $P_{IS,2b}$) according to *Equation 3*:

$$\overline{C}_A = \frac{C_{A-6b} \times P_{IS-6b}}{P_{IS-6b}+P_{IS-5b}+P_{IS-2b}} + \frac{C_{A-5b} \times P_{IS-5b}}{P_{IS-6b}+P_{IS-5b}+P_{IS-2b}} + \frac{C_{A-2b} \times P_{IS-2b}}{P_{IS-6b}+P_{IS-5b}+P_{IS-2b}} \tag{3}$$

On the basis of the $\overline{C}_A$ values of **6a**, **5a**, and **2a** ($\overline{C}_{A,6a}$, $\overline{C}_{A,5a}$ and $\overline{C}_{A,2a}$) in the MeOH extraction of cell lysates, the intracellular concentrations of the analyte **6a**, **5a**, and **2a** ($C_{intra,6a}$ $C_{intra,5a}$ and $C_{intra,2a}$) were calculated according to *Equation 4*. Here $\overline{C}_{A,analyte}$ is the weighted average of the three concentrations in the MeOH extraction, $N$ is the cell number, and $V$ is the mean volume of cells (µL). The mean volume of MDA-MB-231 cells is $1.3 \times 10^{-6}$ µL/cell as reported previously (*Coulter et al., 2012*). The cell number ($N$) was determined with a hemocytometer. The concentration of SAM ($C_{A,SAM}$) in the MeOH extraction was obtained according to *Equation 2*, in which 'X' is the ratio ($P_{A,SAM}/P_{IS,CD3-SAM}$) of the mass peak areas of SAM ($P_{A,SAM}$) versus the internal standard CD$_3$-SAM ($P_{IS,CD3-SAM}$) and 'Y' is the ratio ($C_{A,SAM}/C_{IS,CD3-SAM}$) of the concentrations of SAM versus CD$_3$-SAM in the MeOH extraction. Given the identical LC-MS properties of SAM and CD$_3$-SAM, $C_{A,SAM}$ was obtained solely on the basis of the working curve with CD$_3$-SAM as the internal standard. The intracellular concentration of SAM ($C_{intra,SAM}$) was then calculated using *Equation 4*, in which $N$ is the cell number and $V$ is the mean volume of MDA-MB-231 cells ($1.3 \times 10^{-6}$ µL/cell).

$$C_{inter,\mathbf{analyte}} = \frac{\overline{C}_{A,\mathbf{analyte}} \times 40\mu L}{N \times V} \tag{4}$$

## Quantification of intracellular concentrations of 6b, 5b, 2b and SAM

Similar experimental procedures were carried out to quantify the intracellular concentrations of **6b**, **5b**, **2b** and SAM except that **6b**, **5b**, and **2b** were the analytes and their structurally related **6a**, **5a** and **2a** were used as the internal standards with their $C_{IS}$ values of 0.125 µM, 0.125 µM, and 0.0125 µM, respectively. For each analyte (**6b**, **5b**, and **2b**), three working curves (*Figure 5c,e*, *Figure 5—figure supplement 2*) were generated to cover the concentration range of 3.9 nM–18.0 µM of the analyte ($C_{A,6b}$, $C_{A,5b}$ and $C_{A,2b}$) with **6a**, **5a** and **2a** ($C_{IS-6a}$, $C_{IS-5a}$, and $C_{IS-2a}$) as the LC-MS/MS internal standards, respectively.

## Calculation of the percentages of intracellular apo-CARM1, CARM1 occupied by 6a, 5a, and 2a, and SAM-bonded CARM1 versus the total amount of CARM1

On the basis of the intracellular concentrations ($C_{intra, analyte}$) of **6a**, **5a**, **2a,** and SAM quantified by the LC-MS/MS experiment described above, the percentages of intracellular apo-CARM1, CARM1 occupied by **6a**, **5a**, and **2a**, SAM-bound CARM1 versus the total amount of intracellular CARM1 were calculated using *Equations 5-7*, respectively.

$$\text{Occupancy(Apo)}\% = \frac{[E]}{[E] + [E-SAM] + \sum\limits_{i=1}^{n}[E-I_i]} \times 100\% = \frac{1}{1 + \frac{C_{\text{inter,SAM}}}{K_{m,SAM}} + \sum\limits_{i}^{n} \frac{C_{\text{inter,Ii}}}{K_d \mathbf{Ii}}} \times 100\% \quad (5)$$

$$\text{Occupancy(I)}\% = \frac{\sum\limits_{i=1}^{n}[E-Ii]}{[E] + [E-SAM] + \sum\limits_{i=1}^{n}[E-I_i]} \times 100\% = \frac{\sum\limits_{i}^{n} \frac{C_{\text{inter,Ii}}}{k_d,\mathbf{Ii}}}{1 + \frac{C_{\text{inter,SAM}}}{K_{m,SAM}} + \sum\limits_{i}^{n} \frac{C_{\text{inter,Ii}}}{K_d \mathbf{Ii}}} \times 100\% \quad (6)$$

$$\text{Occupancy(SAM)}\% = \frac{[E-SAM]}{[E] + [E-SAM] + \sum\limits_{i=1}^{n}[E-I_i]} \times 100\% = \frac{\frac{C_{\text{inter,SAM}}}{k_{m,SAM}}}{1 + \frac{C_{\text{inter,SAM}}}{K_{m,SAM}} + \sum\limits_{i}^{n} \frac{C_{\text{inter,Ii}}}{K_d \mathbf{Ii}}} \times 100\% \quad (7)$$

Here [E], [E-SAM] and [E-$I_i$] are the intracellular concentrations of apo-CAMR1, the CARM1–SAM complex and the CARM1 occupied by the inhibitors; $C_{\text{intra,SAM}}$ and $C_{\text{intra,Ii}}$ are the intracellular concentrations of SAM and individual CARM1 inhibitors, respectively; $K_{m,SAM}$ is the Michaelis-Menton constant of SAM in forming the CARM1–SAM complex, which is approximated to $K_{d,SAM}$ ($K_{m,SAM} \approx K_{d,SAM}$ = 245 nM); and $K_{d,Ii}$ is the dissociation constant of these CARM1 inhibitors. In the case of the cellular treatment with **6a**, 'n' is equal to 3; $I_1$, $I_2$ and $I_3$ stand for **2a**, **5a** and **6a**, respectively; $K_{d,I1}$, $K_{d,I2}$ and $K_{d,I3}$ stand for $K_{d,2a}$ = 17 nM, $K_{d,5a}$ = 9 nM and $K_{d,6a}$ = 275 nM.

## Calculation of SMYD2 occupancy by 2a within cells

In a manner similar to that described above for the ligand occupancy of CARM1, the SMYD2 occupancy by **2a** was modeled with $K_{d,SAM}$ = 60 nM, $K_{d,2a}$ = 150 nM and *Equation 6*. Here, the $K_{d,SAM}$ value was reported previously upon developing the activity assay of SMYD2 (*Sweis et al., 2015*). The $K_{d,2a}$ value was obtained according to IC$_{50}$ = [SAM]×$K_d$/$K_{d,SAM}$+$K_d$, in which [SAM]=$K_{d,SAM}$ = 60 nM for the IC$_{50}$ assay and IC$_{50}$ = 0.3 µM was obtained (*Supplementary file 1*-Table A). The terms of SMYD2 occupancy by exogenous ligands were ignored given their low concentrations and low affinity relative to that of **2a** (*Supplementary file 1*-Table A).

## Microsomal stability of 6a

Potential in vivo clearance of **6a** was evaluated in the presence of liver microsomes as previously reported (*Hansen et al., 2012*). Briefly, **6a** was dissolved into 50 mM potassium phosphate buffer (pH 7.4) to a final concentration of 100 µM. To a 0.5 mL Eppendorf vial containing ice-cold potassium phosphate buffer (pH 7.4), we added 5 µL of the 100 µM stock solution of **6a**, 3 µL of a freshly prepared NADPH solution (5 mM, Sigma-Aldrich, N7785), and 1.33 µL of freshly thawed rat liver microsomes (Sigma-Aldrich, M9066) to yield assay samples with a final volume of 50 µL. The resulting mixture was immediately vortexed and incubated in a 37°C shaker. At the time interval of 0, 20, 40, and 60 min in triplicates, respective assay samples were quenched by adding 50 µL of ice-cold methanol, vigorously vortexed, and centrifuged with 5,000 g at 4°C for 20 min. From the methanol-quenched sample, 50 µL of supernatant was collected and mixed with 50 µL of 0.1% (v/v) TFA/MeOH solution containing 0.125 µM **6b**, 0.125 µM **5b**, and 16 µM **2b** as the internal standards. The resulting mixture was transferred into a 96-well plate (Agilent, 5042–1386) and stored at −20°C for LC-MS/MS analysis. The microsomal stability of **6a** was evaluated via LC-MS/MS quantification of the residual **6a** and of two potential microsome-processed products, **5a** and **2a**, upon quantification of intracellular concentrations of **6a**, **5a** and **2a** (as described above). The amount of **6a** was plotted as the percentage of the residual concentration at each time point against its initial concentration at 0 min with GraphPad Prism Software. Potential production of **5a** and **2a** from **6a** was examined in parallel. Interestingly, despite robust accumulation of **5a**, no **2a** was identified with the detection threshold of 0.02 µM.

For negative controls, 3 µL of 50 mM potassium phosphate buffer (pH 7.4) replaced the freshly prepared NADPH solution; two assay samples at the time interval of 0 and 60 min were collected. Under the current microsomal condition, the concentration of **6a** remained the same (10 ± 0.7 µM) and there was no production of **5a** or **2a**.

## Evaluation of methylation marks of CARM1 in breast cancer cells

MDA-MB-231 cells and MCF-7 parental and *CARM1*-KO (*Wang et al., 2014a*) cells were maintained in DMEM (Gibco, Gaithersburg, MD) medium containing 10% FBS (Gibco). These cells were then treated with compounds or with DMSO for 48 hr. The resultant cells were washed twice in phosphate-buffered saline (PBS) and the samples were sonicated in ice-cold RIPA buffer (Thermo, Waltham, MA). The lysates were centrifuged (15,000 g) at 4°C for 15 min. The supernatants were kept with the total protein amount determined with Bradford protein assay (BioRad, Hercules, CA). After quantification, 50 μg of protein from each sample was loaded onto 6% SDS-PAGE and transferred onto a nitrocellulose membrane (PALL, Port Washington, NY). For the Arg1064 methylation mark of BAF155, the blots were blocked in 5% non-fat milk for 1 hr and incubated with anti-me-BAF155 (Cancer Cell, 2014), anti-BAF155 (1:1000; Santa Cruz Biotechnology, Dallas, TX), anti-CARM1 (1:1000; Genemed Synthesis, San Antonio, TX), and anti-β-actin (1:20000; Sigma-Aldrich, St. Louis, MO) overnight at 4°C. After washing three times in Tris-buffered saline with Tween 20 (TBST), the blots were incubated with HRP-conjugated secondary antibody (1:3000; Jackson ImmunoResearch, West Grove, PA). After washing blots with TBST, the membranes were developed using SuperSignal West Pico ECL solution (Thermo). For the Arg455/Arg460 methylation mark of PABP1, anti-me-PABP1 (1:1000) and anti-PABP1 (1:1000) antibodies (Genemed Synthesis, San Antonio, TX) were used instead. Here the antibodies against CARM1, PABP1, and me-PABP1 were custom generated by Genemed Synthesis (San Antonio, TX) (*Zeng et al., 2013*). The density of the protein bands was quantified using ImageJ software (NIH, Bethesda, MD). The $EC_{50}$ values were obtained by fitting the methylation percentage (%) of BAF155 or PABP1 against the concentrations of the inhibitor using a sigmoidal equation with GraphPad Prism Software.

## Cellular thermal shift assay (CETSA)

CETSA was performed as described previously (*Jafari et al., 2014*) to examine the intracellular engagement of **6a** or **6b** with CARM1. Briefly, $2.0 \times 10^6$ MDA-MB-231 cells were incubated with 15 μM **6a** or **6b** and DMSO for 48 hr. The harvested cells were re-suspended in PBS buffer and divided into eight aliquots (30 μL/aliquot), and then heat-shocked at various temperatures (49.1, 54.6, 57.0, 59.5, 61.8, 63.9, 65.6, and 67.0°C) for 3 min with a Bio-Rad CFX96 Real-Time PCR Detection Instrument (using a temperature gradient of 49–67°C). The heat-shocked cells were then lysed with the freeze-thaw method with a liquid nitrogen bath followed by a 25°C water bath for five cycles. The cell lysate was centrifuged at 4°C at 18,000 g for 20 min. The resultant supernatant containing the soluble protein fraction was collected and loaded onto an SDS-PAGE gel (20 μL). Western blotting of CARM1 was performed with anti-CARM1 antibody (Cell Signaling Technology, C31G9). For each sample, the band intensity of CARM1 was quantified with ImageJ. The band intensity of CARM1 was normalized to the band intensity at 49.1°C (the lowest heat-shock temperature). Melting curves were obtained by plotting the normalized band intensity against the heat-shock temperatures and fit with a Boltzmann sigmoidal equation in GraphPad Prism. The melting temperatures ($T_m$) of **6a**, **6b**, or DMSO were obtained as the heat-shock temperatures that correspond to the 50% normalized band intensity in the fitted sigmoidal curves.

## Cell invasion and proliferation assay

MDA-MB-231 parental and *CARM1*-KO cells (*Wang et al., 2014a*) were maintained in DMEM (Gibco, Gaithersburg, MD) medium containing 10% FBS (Gibco). Cell invasion assays were performed using 8.0 μm pore size Transwell inserts (Greiner Bio-One, Kremsmünster, Austria). MDA-MB-231 parental and *CARM1*-KO (Cancer Cell, 2014) cells were harvested with trypsin/EDTA and washed twice with serum-free DMEM (Gibco). $2 \times 10^5$ cells in 0.2 mL serum-free DMEM (Gibco) were seeded onto the upper chamber, which was pre-coated with a thin layer of 40 uL of 2 mg/mL Matrigel (Corning, NY, USA) for 2 hr incubation at 37°C. To the lower chamber, we added 0.6 mL DMEM containing 10% FBS (GIBCO) and compounds or DMSO. After 16 hr in the 37°C incubator, the cells on the inner side of the upper chamber together with the Matrigel layer were removed using cotton tips. The residual invasive cells in the outer side of the upper chamber were fixed in 3.7% formaldehyde (a weight percentage) at an ambient temperature (22°C) for 2 min and 100% methanol for 20 min, and then stained for 15 min with a solution containing 1% crystal violet and 2% ethanol in 100 mM borate buffer (pH 9.0). The number of invasive cells was counted under

a microscope by taking five independent fields. Relative cell invasion was determined by the number of the invasive cells normalized to the total number of cells adhering to 0.8 μm transwell filters. The $EC_{50}$ was obtained with GraphPad Prism Software upon fitting *Equation 8*, in which '%Inhibition' is the percentage of the inhibition of invasiveness, 'Maximal Inhibition%' is the percentage of the maximal inhibition of invasiveness, and [Inhibitor] is the concentration of the inhibitor.

$$\text{Inhibition\%} = \frac{\text{Maximal Inhibition\%} \times [\text{Inhibitor}]}{[\text{Inhibitor}] + \text{EC}_{50}} \qquad (8)$$

To examine proliferation of MDA-MB-231 cells, 5000 cells of parental and *CARM1*-KO cells were seeded in a 96-well plate and incubated in 37°C overnight. These cells were treated with various doses (0.0001–10 μM) of **SKI-73** or **SKI-73N** (**6a** or **6b**) in DMSO and incubated for 72 hr. An MTT assay was then performed to examine viability with DMSO-treated cells as the control. The relative viability of compound-treated cells versus DMSO-treated parent cells were plotted against the concentrations of **SKI-73** and **SKI-73N** (**6a** and **6b**).

## Sample preparation for single-cell RNA-seq analysis (scRNA-seq)

Cell preparation for scRNA-seq. 10 × Genomics droplet-based scRNA-seq was implemented to characterize five types of MDA-MB-231 cells: MDA-MB-231 cells treated with **DMSO**, **SKI-73** (**6a**) and **SKI-73N** (**6b**), MDA-MB-231 cells that freshly invaded through Matrigel, and *CARM1*-KO MDA-MB-231 cells. For the first three samples, MDA-MB-231 cells were treated with DMSO, 10 μM **SKI-73** (**6a**), and 10 μM **SKI-73N** (**6b**) for 48 hr. The resulting cells were trypsinized at 37°C for 3 min. The harvested cells were washed twice with 1 × PBS (phosphate-buffered saline) containing 0.04% (volume%) bovine serum albumin (BSA), gently dispersed to dissociate cells, and then filtered through a cell strainer (100 μm Nylon mesh, Fisherbrand) to obtain single-cell suspensions for scRNA-seq. For the *CARM1*-KO sample, the cells were trypsinized at 37°C for 3 min, washed twice with 1 × PBS containing 0.04% BSA, and filtered through a 100 μm Nylon-mesh cell strainer (Fisherbrand) to obtain single-cell suspensions for scRNA-seq.

To collect the cells that freshly invaded through Matrigel, the conditions described above for cell invasion assays were applied to allow approximately 5% of the $1 \times 10^7$ seeded MDA-MB-231 cells to invade through Matrigel. Briefly, ten Transwell inserts with a diameter of 24 mm and a pore size of 8.0 μm were pre-coated with a thin layer of 54 μL of 2 mg/mL Matrigel (Corning). $1 \times 10^6$ MDA-MB-231 cells in 1.0 mL serum-free DMEM (Gibco) were seeded into the upper chamber of each Matrigel-coated Transwell insert. 2.0 mL DMEM containing 10% FBS (Gibco) was added into the lower chambers. After 16 hr incubation, the cells on the inner side of the upper chamber together with the Matrigel layer were removed using cotton tips. The cells that freshly invaded–those attached on the outer side of the upper chamber–were subjected to 3 min trypsin digestion at 37°C for detachment. The resulting cells were washed twice with 1 × PBS containing 0.04% BSA, gently dispersed to dissociate cells, and filtered through a 100 μm Nylon-mesh cell strainer (Fisherbrand) to obtain single-cell suspensions for scRNA-seq.

## Cell barcoding, library preparation and sequencing

The scRNA-seq libraries were prepared following the user guide manual (CG00052 Rev E) provided by the 10 × Genomics and Chromium Single Cell 3' Reagent Kit (v2). Briefly, samples containing approximately 8700 cells (93–97% viability) were encapsulated in microfluidic droplets at a dilution of 66–70 cells/μL, which resulted in 4369–5457 recovered single-cells per sample with a multiplet rate ~3.9%. The resultant emulsion droplets were then broken and barcoded cDNA was purified with DynaBeads, followed by 12-cycles of PCR-amplification: −98 °C for 180 s, 12×(98°C for 15 s, 67° C for 20 s, 72°C for 60 s), and 72°C for 60 s. The 50 ng of PCR-amplified barcoded-cDNA was fragmented with the reagents provided in the kit and purified by SPRI beads with an averaged fragment size of 600 bp. The DNA library was then ligated to the sequencing adapter followed by indexing PCR: −98 °C for 45 s; 12× (98°C for 20 s, 54°C for 30 s, 72°C for 20 s), and 72°C for 60 s. The resulting DNA library was double-size purified (0.6–0.8×) with SPRI beads and sequenced on Illumina NovaSeq platform (R1 – 26 cycles, i7 – eight cycles, R2 – 96 cycles) resulting in 70–79 million reads per sample with average reads per single-cell being 8075–10,342 and average reads per transcript 1.11–1.15.

## Processing, transformation, filtering and dimensionality reduction of scRNA-seq data

The fastq files containing the transcriptome and barcoding metadata were demultiplexed using the SEquence Quality Control (SEQC) pipeline (http:github.com/ambrosejcarr/seqc.git) resulting in around 8000 UMIs per one cell. The table of UMI counts was used as the input and Seurat package v.2.3.4 (*Butler et al., 2018*) was applied for scRNA-seq analysis. Here, the raw UMI counts were normalized per cell by dividing the total number of UMIs in each individual cell, multiplying by a scale factor of 10,000 and transforming into natural logarithm values. Cells with 1000~5000 genes and <20% mitochondrial RNA transcripts were kept for further analysis. Dimensionality reduction was carried out by selecting a set of highly variable genes on the basis of the average expression and dispersion per gene. The set of genes was used for principle component analysis (PCA). Top principle components were then chosen for cell clustering analysis and *t*-SNE projection. Regression was performed to remove cell–cell variation in gene expression driven by the UMI number, mitochondrial gene content and ribosomal gene content using 'ScaleData' function in Seurat package (*Nestorowa et al., 2016*). Clusters of cells were identified by clustering algorithm based on a shared nearest neighbor (SNN) modularity optimization that was included in Seurat package v.2.3.4 (*Butler et al., 2018*).

## Subpopulation clustering guided by scRNA-seq

### Cell-cycle awareness

To assign cells to cell-cycle stages ($G_0/G_1$, S and $G_2$/M), individual cells were scored on the basis of their expression of $G_2$/M-phase and S-phase markers (*Nestorowa et al., 2016*) by comparing the average expression of these markers with that of a random set of background genes (*Tirosh et al., 2016*). Cells with positive higher S-phase or $G_2$/M-phase scores were assigned as S-phase or $G_2$/M-phase cells, respectively. Cells with negative S-phase and $G_2$/M-phase scores were assigned as non-S/$G_2$/M-phase cells and annotated as $G_0/G_1$-phase cells. The whole cell population, as well as its subpopulations, can thus be classified into the three groups according to their cell-cycle scores.

### Determination of the number of clusters

Three algorithms–Silhouette analysis, entropy scoring, and Fisher's Exact Test–were applied collectively to determine the number of clusters.

### Silhouette analysis

Silhouette analysis was conducted with no awareness of cell origins (DMSO-, **SKI-73**(**6a**)- or **SKI-73N** (**6b**)-treated cells) and was calculated on the basis of distances defined as Euclidean distance between any pair of cells on the two-dimensional *t*-SNE projection (*de Amorim and Hennig, 2015*; *Rousseeuw, 1987*).

### Entropy scoring

We developed an entropy-based scoring method to evaluate the efficiency of clustering subpopulations with the three cell origins (DMSO-, **SKI-73**(**6a**)-, and **SKI-73N**(**6b**)-treated). Entropy score is defined within a range of 0–1 by *Equation 9*, which was derived on the basis of the double-weighted sum of cell-origin-based Shannon entropy across clustered subpopulations. Herein, '*i*' is the series number of a clustered subpopulation starting from zero; '*n*' is the largest series number of clustered subpopulations; the total number of subpopulation is '*n+1*' (*i* = 0–n); '$f_i$' corresponds to the fraction of the subpopulation '*i*' in the total cell population ($0 < f_i \leq 1; \Sigma_{i=1}^{n} f_i = 1$) ; '*j*' represents one of the three cell origins (*j* = 1, 2 or 3); '$d_{j,i}$' ('$d_{DMSO,i}$', '$d_{SKI-73N,i}$' and '$d_{SKI-73,i}$') is the fractional distribution of the cells with the '*j*' origin (DMSO, **SKI-73/6a**, and **SKI-73N/6b**) within the '*i*' subpopulation ($0 \leq d_{j,i} \leq 1; -\Sigma_{j=1}^{3} d_{j,i} = 1$) ; '$f_j$' is the fraction of the cells with the '*j*' origin within the total population ($0 < f_i \leq 1; \Sigma_{j=1}^{3} f_i = 1$) ; '$-\Sigma_{j=1}^{3} (f_i \times \log_e(f_j))$' is the theoretical maximum of ($-\Sigma_{i=1}^{0} (f_i \times (\Sigma_{j=1}^{3} (d_{j,i} \times \log_e(d_{j,i})))))$)(all the cells in a single cluster). A smaller entropy score indicates that the corresponding method allows cell subpopulations to be clustered with higher resolution for the DMSO-, **SKI-73**(**6a**)-, and **SKI-73N**(**6b**)-treated cells. The minimal entropy score of zero indicates that subpopulations can be fully resolved for the three treatment conditions.

$$\text{Entropy Score} = \left(-\Sigma_{i=0}^{n}(f_i \times (\Sigma_{j=1}^{3}(d_{j,i} \times \log_e(d_{j,i}))))\right)/\left(-\Sigma_{j=1}^{3}(f_j \times \log_e(f_i))\right) \quad (9)$$

## Fisher's Exact Test

Fisher's Exact Test was implemented to evaluate the agreement of the clusters with the three cell origins (DMSO-, **SKI-73**(**6a**)- or **SKI-73N**(**6b**)-treated cells) using R package 'fisher.test': http://mathworld.wolfram.com/FishersExactTest.html (**Mehta and Patel, 1983**). Because of the significant computation cost of Fisher's Exact Test, the cell population of each Fisher's Exact Test was downsampled to 150 cells and this process was repeated for 100 times to cover a majority of the cell population. The p-value of Fisher's Exact Test was then computed by Monte Carlo simulation. Means and standard errors of p-values were calculated and reported as the outputs of Fisher's Exact Test.

Three algorithmic scoring systems were used over a range of the resolution parameter that sets the corresponding 'granularity' of clustering, with higher values indicating a greater number of clusters. Here, silhouette analysis was applied to determine the number of clusters in an unsupervised manner, without awareness of the three cell origins (DMSO-, **SKI-73**(**6a**)- or **SKI-73N**(**6b**)-treated cells). The entropy scoring and Fisher's Exact Test were implemented to evaluate the biological meaning of the clustering, using the minimal cluster number to resolve the cells between the three treatment conditions maximally. Given the awareness of the three cell origins, the minimal number of clusters with the maximal resolution of cell origin guided by the entropy scoring and Fisher's Exact Test is expected within the 1~3-fold range of the optimized number of clusters guided by Silhouette analysis. Fisher's Exact Test was used as the primary scoring method to determine the efficiency of clustering, given its higher resolution.

## Population analysis of the three treatment conditions

'$d_{j,i}$' ('$d_{DMSO,i}$', '$d_{SKI-73N,i}$' and '$d_{SKI-73,i}$' in **Equation 9** are defined as the fraction of the cells with the '$j$' origin ($j$ = 1, 2 or three for the treatment with DMSO, **SKI-73N/6b** and **SKI-73/6a**, respectively) within the '$i$' subpopulation ($i$ = 0–n for the clustered subpopulation). '$d_{i,total}$' is defined as the fraction of the cells of the '$i$' subpopulation within the total cell population in each cell-cycle stage. '$d_{(j,i),total} = d_{j,i} \times d_{i,total}$' represents the fraction of the cells with the '$j$' origin (DMSO, **SKI-73/6a**, and **SKI-73N/6b**) and the '$i$' subpopulation within the total cell population in each cell-cycle stage. For a specific subpopulation '$i$', there are three '$d_{(j,i),total}$' values ($d_{(DMSO,i),total}$, $d_{(SKI-73N,i),total}$ and $d_{(SKI-73,i),total}$) for the treatments with DMSO, **SKI-73N** (**6b**) and **SKI-73** (**6a**), respectively. The alteration of subpopulations is defined as the ratios of $d_{(SKI-73N,i),total}/d_{(DMSO,i),total}$ or $d_{(SKI-73,i),total}/d_{(DMSO,i),total}$ fall out of the range of 1.0 ± 0.2. Population analysis was conducted by classifying the **SKI-73N/SKI-73** (**6a/6b**)-treated subpopulations into the following five categories: commonly resistant (0.8 < '$d_{(SKI-73N,i),total}/d_{(DMSO,i),total}$' and '$d_{(SKI-73,i),total}/d_{(DMSO,i),total}$'<1.2), commonly emerging ('$d_{(SKI-73N,i),total}/d_{(DMSO,i),total}$' and '$d_{(SKI-73,i),total}/d_{(DMSO,i),total}$'≥1.2), commonly depleted ('$d_{(SKI-73N,i),total}/d_{(DMSO,i),total}$' and '$d_{(SKI-73,i),total}/d_{(DMSO,i),total}$'≤0.8; |'$d_{(SKI-73N,i),total}/d_{(DMSO,i),total}$'−'$d_{(SKI-73,i),total}/d_{(DMSO,i),total}$'|<0.15), differentially emerging (either '$d_{(SKI-73N,i),total}/d_{(DMSO,i),total}$' or '$d_{(SKI-73,i),total}/d_{(DMSO,i),total}$'>1.2), and differentially depleted ('$d_{(SKI-73N,i),total}/d_{(DMSO,i),total}$' or '$d_{(SKI-73,i),total}/d_{(DMSO,i),total}$'<0.8; |'$d_{(SKI-73N,i),total}/d_{(DMSO,i),total}$'−'$d_{(SKI-73,i),total}/d_{(DMSO,i),total}$'|>0.15). Although additional combinations of differentially altered subpopulations could be possible, only those defined above were found under our treatment conditions.

## Correlation analysis of subpopulations

In each cell cycle ($G_0/G_1$, S and $G_2/M$) of the cells treated with DMSO, **SKI-73** (**6a**) or **SKI-73N** (**6b**) and 'invasion cells', correlation analysis of subpopulations was conducted with 'BuildClusterTree' function in the Seurat package (https://rdrr.io/cran/Seurat/man/BuildClusterTree.html) (**Nestorowa et al., 2016**). The phylogenetic trees were constructed by averaging gene expressions across all cells in each subpopulation and then calculating distance on the basis of expressions averaged between different subpopulations.

## Differential expression across remotely related subpopulations and the selection of representative transcripts

Differentially expressed genes were identified by comparing two groups of cells using the Wilcox rank sumtest with the 'FindMarkers' function in the Seurat package (**Nestorowa et al., 2016**). In

particular, the 'invasion cells' and their most correlated clusters (which were revealed in the correlation analysis) were selected as the 'high' group; the remaining remotely related clusters were selected as the 'low' group. The differential expression analysis was then performed by comparing cells in the 'high' group with the cells in the 'low' group. For the $G_0/G_1$-phase cells, the 'invasion cells' and Subpopulation 6, 7, 8, 9, 14 were selected as the 'high' group; Subpopulation 0, 1, 2, 3, 4, 5, 10, 11, 12, 13, 15, 16, 17, 18, 19, 20 were selected as the 'low' group. For the $G_2/M$-phase cells, the 'invasion cells' and Subpopulations 1, 2 were selected as the 'high' group; and Subpopulations 0, 3, 4, 5 were selected as the 'low' group. For the S-phase cells, the 'invasion cells' and Subpopulations 0, 3 were selected as the 'high' group; and Subpopulations 1, 2, 4, 5, 6 were selected as the 'low' group. Differentially expressed genes were ranked according to the average $\log_2$ fold change 'avg_logFC' and the adjusted p-values 'p_val_adj' with the Seurat package. Top upregulated and downregulated genes were chosen by setting a cutoff on their 'avg_logFC' values ($>0.25$ or $<-0.25$). Then, curated genes with potential functional relevance to cancer malignancy (30 upregulated and 10 downregulated genes) were selected as representative genes for generating heat map plots using the 'DoHeatmap' function in the Seurat package.

## Analysis of differential expression across invasion-prone subpopulation candidates and the selection of representative transcripts for Violin plots

To select candidate genes of the $G_0/G1$-phase cells for violin plots, the 'invasion cells' and the cells of Subpopulation 8 were selected as the 'high' group and the Subpopulations 6, 7, 9, 14 were selected as the 'low' group. Differentially expressed genes were ranked according to the average $\log_2$ fold change 'avg_logFC' and adjusted p-values 'p_val_adj'. Top up- and downregulated genes were chosen by setting their 'avg_logFC' values $> 0.25$ or $< -0.25$ and by curating genes that are functionally implicated in cancer malignancy. Heat map plots were generated for the selected gene with the 'DoHeatmap' function in the Seurat package (Nestorowa et al., 2016). Furthermore, a panel of the top eight genes highlighting similarity between the invasion-prone Subpopulation 8 and 'invasion cells' (the top five upregulated and top three downregulated genes) were selected for generating violin plots.

## Acknowledgements

The authors thank Christina Leslie for suggesting scRNA-seq analysis. Funding has been provided by the US National Institutes of Health (ML: R01GM096056, R01GM120570, R35GM131858), the US National Cancer Institute (ML: 5P30 CA008748; WX: R01CA236356, R01CA213293; LXQ: CA214845, CA008748), the Starr Cancer Consortium (ML), the MSKCC Functional Genomics Initiative (ML), the Sloan Kettering Institute (ML), the Mr. William H Goodwin and Mrs. Alice Goodwin Commonwealth Foundation for Cancer Research, the Experimental Therapeutics Center of Memorial Sloan Kettering Cancer Center (ML), the MSKCC Metastasis and Tumor Ecosystems Center (ML), the Tri-Institutional PhD Program in Chemical Biology (SC), the Susan G Komen Foundation (EJK: PDF17481306), and Special Funding of Beijing Municipal Administration of Hospitals Clinical Medicine Development (YangFan Project) (ZZ: ZYLX201713). The Structural Genomics Consortium is a registered charity (no. 1097737) that receives funds from AbbVie, Bayer Pharma AG, Boehringer Ingelheim, the Canada Foundation for Innovation, the Eshelman Institute for Innovation, Genome Canada, the Innovative Medicines Initiative (EU/EFPIA) (ULTRA-DD grant no. 115766); Janssen, Merck KGaA (Darmstadt, Germany), MSD, Novartis Pharma AG, the Ontario Ministry of Economic Development and Innovation, Pfizer, the São Paulo Research Foundation-FAPESP, Takeda, and the Wellcome Trust. The X-ray structure results for CARM1 are derived from work conducted at the Northeastern Collaborative Access Team beamlines, which are funded by the National Institute of General Medical Sciences from the National Institutes of Health (P30 GM124165). The Eiger 16M detector on the 24-ID-E (NE-CAT) beam line is funded by a NIH-ORIP HEI grant (S10OD021527). This research used resources of the Advanced Photon Source, a U.S. Department of Energy (DOE) Office of Science User Facility operated for the DOE Office of Science by Argonne National Laboratory under Contract No. DE-AC02-06CH11357. Additional work was performed using beamline 23-ID-B (GM/CA). GM/CA@APS has been funded in whole or in part with Federal funds from the

National Cancer Institute (ACB-12002) and the National Institute of General Medical Sciences (AGM-12006). The Eiger 16M detector at GM/CA-XSD was funded by NIH grant S10 OD012289..
   Dr. Zhang unfortunately passed away during the revision of this manuscript.

## Additional information

### Funding

| Funder | Grant reference number | Author |
| --- | --- | --- |
| National Institutes of Health | R01GM096056 | Minkui Luo |
| National Institutes of Health | R01GM120570 | Minkui Luo |
| National Cancer Institute | 5P30 CA008748 | Minkui Luo |
| National Cancer Institute | R01CA236356 | Wei Xu |
| National Cancer Institute | R01CA213293 | Wei Xu |
| Starr Cancer Consortium | I8-A8-058 | Minkui Luo |
| Memorial Sloan Kettering Cancer Center | Functional Genomics Initiative | Minkui Luo |
| Mr William H Goodwin and Mrs Alice Goodwin Commonwealth Foundation for Cancer Research | | Minkui Luo |
| Memorial Sloan Kettering Cancer Center | Metastasis and Tumor Ecosystems Center | Minkui Luo |
| Susan G. Komen | PDF17481306 | Eui-jun Kim |
| Beijing Municipal Administration of Hospitals Clinical Medicine Development of Special Funding Support | YangFan Project - ZYLX201713 | Zhenyu Zhang |
| The Structural Genomics Consortium | | Peter J Brown |
| National Institutes of Health | R35GM131858 | Minkui Luo |
| Memorial Sloan Kettering Cancer Center | Experimental Therapeutics Center | Minkui Luo |
| National Cancer Institute | CA214845 | Li-Xuan Qin |
| National Cancer Institute | CA008748 | Li-Xuan Qin |
| Sloan Kettering Institute | | Minkui Luo |
| Tri-institutional PhD Program in Chemical Biology | | Shi Chen |

The funders had no role in study design, data collection and interpretation, or the decision to submit the work for publication.

### Author contributions

Xiao-Chuan Cai, Conceptualization, Resources, Data curation, Formal analysis, Supervision, Validation, Investigation, Visualization, Methodology, Writing—original draft, Project administration; Tuo Zhang, Conceptualization, Resources, Data curation, Software, Formal analysis, Investigation, Visualization, Methodology; Eui-jun Kim, Formal analysis, Funding acquisition, Validation, Investigation, Methodology; Ming Jiang, Junyi Wang, Nawei Zhang, Hong Wu, Fengling Li, Carlo C dela Seña, Hong Zeng, Validation, Investigation; Ke Wang, Resources, Validation, Investigation, Methodology; Shi Chen, Investigation, Visualization, Methodology; Victor Vivcharuk, Validation, Investigation, Methodology; Xiang Niu, Conceptualization; Weihong Zheng, Linas Mazutis, Investigation, Methodology; Jonghan P Lee, Yuling Chen, Dalia Barsyte, Magda Szewczyk, Taraneh Hajian, Glorymar Ibáñez, Aiping Dong, Ludmila Dombrovski, Investigation; Zhenyu Zhang, Conceptualization, Funding

acquisition; Haiteng Deng, Jinrong Min, Conceptualization, Resources, Funding acquisition; Cheryl H Arrowsmith, Supervision, Funding acquisition; Lei Shi, Conceptualization, Resources, Supervision, Methodology; Masoud Vedadi, Supervision, Validation, Investigation; Peter J Brown, Jenny Xiang, Resources, Supervision, Funding acquisition; Li-Xuan Qin, Conceptualization, Formal analysis, Supervision, Funding acquisition, Methodology; Wei Xu, Conceptualization, Formal analysis, Supervision, Funding acquisition; Minkui Luo, Conceptualization, Data curation, Formal analysis, Supervision, Funding acquisition, Validation, Investigation, Visualization, Methodology, Writing—original draft, Project administration

### Author ORCIDs
Shi Chen (iD) http://orcid.org/0000-0002-5860-2616
Peter J Brown (iD) http://orcid.org/0000-0002-8454-0367
Minkui Luo (iD) https://orcid.org/0000-0001-7409-7034

### Decision letter and Author response
Decision letter https://doi.org/10.7554/eLife.47110.sa1
Author response https://doi.org/10.7554/eLife.47110.sa2

# Additional files

### Supplementary files
• Supplementary file 1. Tables A–L. (A) IC$_{50}$ values of SAH, SNF, **1**, **2a**, **2b**, **5a** and **5b** against CARM1. (B–D) Structural analysis of CARM1 in complex with **1**, SNF and SAH. (E–L) Analysis of single-cell RNA-seq data.

• Transparent reporting form

### Data availability
The crystallographic coordinates and structural factors are deposited into the Protein Data Bank with the accession numbers of 4IKP for the CARM1-1 complex and 6D2L for CARM1-5a complex.

The following datasets were generated:

| Author(s) | Year | Dataset title | Dataset URL | Database and Identifier |
| --- | --- | --- | --- | --- |
| Dong A, Dombrovski L, He H, Ibanez G, Wernimont A, Zheng W, Bountra C, Arrowsmith CH, Edwards AM, Brown PJ, Min J, Luo M, Wu H, Structural Genomics Consortium (SGC) | 2013 | Crystal structure of coactivator-associated arginine methyltransferase 1 with methylenesinefungin | https://www.rcsb.org/structure/4IKP | Protein Data Bank, 4IKP |
| Dong A, Zeng H, Walker JR, Hutchinson A, Seitova A, Luo M, Cai XC, Ke W, Wang J, Shi C, Zheng W, Lee JP, Ibanez G, Bountra C, Arrowsmith CH, Edwards AM, Brown PJ, Wu H, Structural Genomics Consortium (SGC) | 2018 | Crystal structure of human CARM1 with (S)-SKI-72 | https://www.rcsb.org/structure/6D2L | Protein Data Bank, 6D2L |

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
