## [Decision Letter]

**Acceptance summary:**

This manuscript describes the development of the small molecular weight compound SKI-73 as a prodrug that releases two potent and selective inhibitors of protein arginine methyltransferase 4 (CARM1) in cells. Using a previously reported inhibitor from natural sources as a scaffold, the authors developed the two inhibitors that display sub-nanomolar potency against the target enzyme and >10-fold selectivity over 7 other human protein arginine methyltransferase and 26 methyltransferases of other classes. Two other chemical probes for the target enzyme have been identified previously. Importantly, all three of these chemical probes have distinct molecular scaffolds and mechanisms. The current compounds compete for the cofactor binding site, unlike the previous two inhibitors that target the substrate arginine binding pocket. The authors next added chemical groups to a primary amine of the initial non-cell permeable inhibitor to develop the cell-permeable pro-drug SKI-73. This strategy potentially delivers a new class of compounds for inhibiting methyltransferases. Detailed single-cell transcription analysis revealed that SKI-73 alters epigenetic plasticity and that the subpopulation of cells that is reduced upon SKI-73 treatment is similar to that of freshly isolated invasive cells. The experimental approach used to arrive at this conclusion will be applicable to studies involving other epigenetic tool compounds. Overall this is an excellent study on the development of CARM1-selective chemical probe with potential for the treatment of metastatic breast cancer.

**Decision letter after peer review:**

Thank you for submitting your article "A chemical probe of CARM1 alters epigenetic plasticity against breast cancer cell invasion" for consideration by *eLife*. Your article has been reviewed by three peer reviewers, and the evaluation has been overseen by Wilfred van der Donk as Reviewing Editor and Philip Cole as the Senior Editor. The following individuals involved in review of your submission have agreed to reveal their identity: Paul R Thompson; Mark Bedford.

The reviewers have discussed the reviews with one another and the Reviewing Editor has drafted this decision to help you prepare a revised submission.

Summary:

This manuscript describes the development of SKI-73 as a prodrug that releases two potent and selective inhibitors of protein arginine methyltransferase 4 (PRMT4 or CARM1) in cells. Using 6'-homosinefungin (HSF) as a core scaffold, the authors developed two inhibitors (2a and 5a) with sub-nanomolar potency for CARM1 and >10-fold selectivity over 7 human PRMTs and 26 methyltransferases of other classes. A related compound that lacks a 6'-methylene group (5b) exhibited less activity. Two other CARM1 chemical probes have been identified previously – EZM2302 and TP-064. Importantly, all three of these chemical probes have distinct molecular scaffolds. The current study shows that compounds 2a and 5a compete for the SAM binding site, unlike the other two inhibitors that target the substrate arginine binding pocket. The authors masked the primary amine of 5a to develop SKI-73, a cell permeable pro-drug. Surprisingly, SKI-73 generated the parent inhibitor 5a in MDA-MB-231 cells along with 2a and these compounds accumulated to relatively high concentrations. They potently inhibited methylation of BAF155 in MCF-7 and MDA-MB-231 cells and PABP1 in MDA-MB-231 cells, two well-known CARM1 substrates. Furthermore, SKI-73 efficiently blocked invasion of MDA-MB-231 without affecting proliferation. This strategy potentially delivers a new class of compounds for inhibiting methyltransferases. Detailed single-cell RNA-seq analysis revealed that SKI-73 alters epigenetic plasticity and that the subpopulation of cells that is lost/reduced upon SKI-73 treatment is similar to that of freshly isolated invasive cells. This approach will be applicable to studies involving other epigenetic tool compounds. Overall this is an excellent study on the development of CARM1-selective chemical probe with potential for the treatment of metastatic breast cancer. The work is potentially suitable for publication in *eLife* after the authors address a number of points raised during review.

Essential revisions:

1) The authors should determine the stability of 6a in microsomes to better account for the formation of 2a from 6a in cells.

2) In subsection “Characterization of 6a (SKI-73) as a chemical probe of CARM1” the authors mention that 6b, the negative control, also is processed to 5b and 2b. Therefore, the authors should provide information on the selectivity of 2b in Figure 1. Only data is shown for 5b, which appears quite selective and has a relatively low IC_50_. The assumption in the discussion is that 2b and 5b will not bind (the authors state that they interact poorly with CARM1), but this is not actually shown.

3) Compound 2a seems to be a fairly good inhibitor of SMYD2 (Figure 1C). When SKI-73 is taken up by cells and processed, it primarily results in the accumulation of 2a (Figure 5—figure supplement 6). Thus, part of the phenotype seen in the epigenetic reprogramming studies may be due to SMYD2 inhibition. This issue needs to be addressed in the following way:

a) Are SMYD2 methylation sites impacted by the treatment of SKI-73? For this experiment, methyl-specific antibodies to known non-histone substrates (like p53, Rb or HSP90) should be tested;

b) Tagged SMYD2 (or endogenous) should be tested in the CETSA assay as performed in Figure 4F;

c) The issue of potential SKI-73 off-target effects should be addressed in the discussion.

4) Figure 6C shows a major shift in the subpopulation of cells that display SKI-73 specific depletion. This finding is nicely linked to the subpopulation of freshly isolated invasive cells. The other dramatic change is in the "SKI-73 specific emerging" cell population (3,4,5,6,16). Is there anything unique about these subpopulations of cells?

5) In general, the medicinal chemistry and compound design are not clearly described and this part of the paper is not easy to follow. The authors should be able to be explain the reasoning and design in a clearer manner. Similarly, the details of the biochemical and cellular assays should be better described and discussed.

---

## [Author Response]

Essential revisions:1) The authors should determine the stability of 6a in microsomes to better account for the formation of 2a from 6a in cells.

Thanks, reviewers, for this excellent suggestion. We have conducted this experiment (Figure 5—figure supplement 5, Materials and methods). While we can observe the consumption of 6a and the production of 5a in the presence of microsomes, the production of 2a could not be observed. This result suggests that the activity to produce 2a is absent in microsomes but present in breast cancer cells. We have included and discussed these results in the third paragraph of the session “Characterization of 6a (SKI-73) as a chemical probe of CARM1”.

2) In subsection “Characterization of 6a (SKI-73) as a chemical probe of CARM1” the authors mention that 6b, the negative control, also is processed to 5b and 2b. Therefore, the authors should provide information on the selectivity of 2b in Figure 1. Only data is shown for 5b, which appears quite selective and has a relatively low IC_50_. The assumption in the discussion is that 2b and 5b will not bind (the authors state that they interact poorly with CARM1), but this is not actually shown.

We apologized for this ignorance. Some of the IC_50_ values including the IC_50_ against CARM1 were reported by us (Zheng et al., 2012). We thus measured the rest IC_50_ and combined these data in Figure 1C and Supplementary file—table S1.

3) Compound 2a seems to be a fairly good inhibitor of SMYD2 (Figure 1C). When SKI-73 is taken up by cells and processed, it primarily results in the accumulation of 2a (Figure 5—figure supplement 6). Thus, part of the phenotype seen in the epigenetic reprogramming studies may be due to SMYD2 inhibition. This issue needs to be addressed in the following way:a) Are SMYD2 methylation sites impacted by the treatment of SKI-73? For this experiment, methyl-specific antibodies to known non-histone substrates (like p53, Rb or HSP90) should be tested;b) Tagged SMYD2 (or endogenous) should be tested in the CETSA assay as performed in Figure 4F;c) The issue of potential SKI-73 off-target effects should be addressed in the discussion.

Thanks to the reviewer for this excellent point. We noted that, while 2a’s IC_50_ against SMYD2 is only 10-fold lower that against CARM1, K_d_ value of SMYD2 to bind SAM is 4-fold lower than that of CARM1. The combined effect leads to 37-fold larger K_d,2a_/K_d,SAM_ ratio of SMYD2 relative to that of CARM1. We argue that the K_d,2a_/K_d,SAM_ ratios are more relevant to evaluate the selectivity and 2a inhibits SMYD2 at much higher concentrations than CARM1. When we planed the experiments to directly examine SMYD2-mediated methylation as suggested by reviewers, we encountered the challenge because the reported experiments were conducted by overexpressing SMYD2 targets in the presence of high-level or overexpressed SMYD2 (Sweis et al., 2015). Despite this difficulty, for the methylation depletion assay as well as the CETSA assay, we agree that 2a might show the off-target effect at extremely high concentrations. To evaluate whether the concentrations of 2a used in our cellular assays are sufficient to inhibit SMYD2, we modeled the occupancy of 2a on SMYD2 in a similar manner as described for 2a’s occupancy on CARM1 (Figure 5E versus Figure 5—figure supplement 4). At the EC_50_ (1.3 µM) against cancer cell invasion (Figure 6A), 2a’s occupancy on SMDY2 is only around 10%. We have included and discussed these points in the second paragraph of the session “Characterization of 6a (SKI-73) as a chemical probe of CARM1”.

4) Figure 6C shows a major shift in the subpopulation of cells that display SKI-73 specific depletion. This finding is nicely linked to the subpopulation of freshly isolated invasive cells. The other dramatic change is in the "SKI-73 specific emerging" cell population (3,4,5,6,16). Is there anything unique about these subpopulations of cells?

Thanks to the reviewer for this suggestion. We further conducted population analysis and found that Subpopulations 4 and 16 account for 90% of the emerging subset. The transcriptional signatures (likely the associated invasion capability) of Subpopulations 4 and 16 are dramatically different from those of the freshly harvested invasive cells and the bulk population of the parental cells including the invasion-prone Subpopulation-8 (Figure 7B,D). SKI-73-specific emerging subpopulations are expected to be suppressed by CARM1 but emerge upon its inhibition, which relate to general biology of CARM1. We have included this discussion in the first paragraph of the session “CARM1-associated epigenetic plasticity of breast cancer cells with single-cell resolution”; the second paragraph of the session “Identification of CARM1-dependent, invasion-prone subpopulations of breast cancer cells”.

5) In general, the medicinal chemistry and compound design are not clearly described and this part of the paper is not easy to follow. The authors should be able to be explain the reasoning and design in a clearer manner. Similarly, the details of the biochemical and cellular assays should be better described and discussed.

Thanks to the reviewer for this concern. We have included more details/rationales of medicinal chemistry and assays in the sessions of “Development of 6′-homosinefungin derivatives as potent and selective CARM1 inhibitors” and “Modes of interaction of 6′-homosinefungin derivatives as CARM1 inhibitors”.